# A magneto-activated nanoscale cytometry platform for molecular profiling of small extracellular vesicles

Kangfu Chen [1,9], Bill T. V. Duong[2,9], Sharif U. Ahmed[1], Piriththiv Dhavarasa[3], Zongjie Wang [4], Mahmoud Labib [1,5,6], Connor Flynn [2,5], Jingya Xu[2], Yi Y. Zhang[1], Hansen Wang [1], Xiaolong Yang[1], Jagotamoy Das [5], Hossein Zargartalebi[1], Yuan Ma[1] & Shana O. Kelley [1,2,3,4,5,7,8] ✉

Exosomal PD-L1 (exoPD-L1) has recently received significant attention as a biomarker predicting immunotherapeutic responses involving the PD1/PD-L1 pathway. However, current technologies for exosomal analysis rely primarily on bulk measurements that do not consider the heterogeneity found within exosomal subpopulations. Here, we present a nanoscale cytometry platform NanoEPIC, enabling phenotypic sorting and exoPD-L1 profiling from blood plasma. We highlight the efficacy of NanoEPIC in monitoring anti-PD-1 immunotherapy through the interrogation of exoPD-L1. NanoEPIC generates signature exoPD-L1 patterns in responders and non-responders. In mice treated with PD1-targeted immunotherapy, exoPD-L1 is correlated with tumor growth, PD-L1 burden in tumors, and the immune suppression of CD8+ tumor-infiltrating lymphocytes. Small extracellular vesicles (sEVs) with different PD-L1 expression levels display distinctive inhibitory effects on CD8 + T cells. NanoEPIC offers robust, high-throughput profiling of exosomal markers, enabling sEV subpopulation analysis. This platform holds the potential for enhanced cancer screening, personalized treatment, and therapeutic response monitoring.

The discovery of the immune checkpoint PD-L1 has revolutionized cancer therapy. Interventions disrupting the PD-L1/PD-1 axis have improved clinical outcomes in numerous cancers including kidney, lung, breast, colon, and melanoma. However, due to the heterogeneous nature of tumors and the complexity of immunoregulatory mechanisms, only a subset of patients respond to immunotherapy[1,2]. Therefore, diagnostic tools that can stratify patients and enable the early classification of responders and non-responders would allow for earlier decision-making and maximize the success of cancer treatments[3]. While cell surface PD-L1 (cPD-L1) is one of the most validated predictive biomarkers for immunotherapy, it has only been shown to be predictive in less than 30% of patients[3,4].

Recent studies have shown that small extracellular vesicles (sEVs) can present PD-L1 on their surfaces and act as improved informative biomarkers compared to cPD-L1[5–8]. sEVs are a subset of extracellular vesicles (EV) which are lipid-bilayer enclosed structures secreted by cells, and exosomes are the most studied sEVs. With a typical size range of 30–160 nm, sEVs can carry molecular constituents such as proteins,

[1]Department of Pharmaceutical Sciences, University of Toronto, Toronto, ON, Canada. [2]Department of Chemistry, University of Toronto, Toronto, ON, Canada. [3]Department of Biochemistry, University of Toronto, Toronto, ON, Canada. [4]Department of Biomedical Engineering, Northwestern University, Evanston, IL, USA. [5]Department of Chemistry, Northwestern University, Evanston, IL, USA. [6]Peninsula Medical School, Faculty of Health, University of Plymouth, Plymouth, UK. [7]Institute for Biomaterials and Biomedical Engineering, University of Toronto, Toronto, ON, Canada. [8]Chan Zuckerberg Biohub Chicago, Chicago, IL, USA. [9]These authors contributed equally: Kangfu Chen, Bill T. V. Duong. ✉e-mail: shana.kelley@northwestern.edu

metabolites, lipids, and nucleic acids from parent cells and act as communication cargos in the tumor microenvironment[9,10]. Due to their high stability and high abundance in bodily fluids such as blood, urine, and saliva, sEVs are promising cancer biomarkers[10-13]. Exosomal PD-L1 (exoPD-L1) in particular has been shown to be a robust indicator of tumor progression and predictor of immunotherapeutic response[5-8,14-19]. Despite the importance of analyzing exosomal markers, there are many technical limitations in this area due to the small size and heterogeneity of tumor-derived sEVs[20-22]. Recent studies indicate that PD-L1 expression among EVs can be heterogeneous and that these EVs show PD-L1-dependent inhibition of T cell activation[23].

Conventional approaches for isolating sEVs such as ultra-centrifugation, filtration, and polymer precipitation are time-consuming and yield low recovery and purity[24,25]. Additionally, standard methods for sEV analysis, such as Western blot and enzyme-linked immunosorbent assays (ELISA), are limited by their relatively low sensitivity and large sample requirements[26]. Several studies have attempted exosomal protein screening using flow cytometry[27,28]. In fact, with optimized high-end cytometers, PD-L1 positive EVs have been detected in clinical samples and shown relevance to immunotherapy outcome[18,19]. However, the detection of EVs with a size smaller than 100 nm is still a challenge and the throughput needs to be improved. More recent studies have applied novel nanomaterials[29,30] or microfluidic systems to detect sEVs and profile their protein contents[31-38]. These efforts have led to several platforms that are capable of multiparametric analysis of sEVs in clinical specimens, such as nanoplasmonic sensors[39,40], electrochemical sensors[41], and immunomagnetic approaches[42]. Despite their excellent analytical performance and broad diagnostic utilities, these methods also lack the throughput needed to handle large volumes of plasma samples. Although our understanding of the translational potential of sEVs is quickly advancing, our ability to further characterize their biological roles in diseases and exploit them as clinical biomarkers is greatly hindered without techniques to efficiently separate sEVs based on their heterogeneous marker expression.

In recent years, our group has developed a number of immunoaffinity-based microfluidic technologies for the rapid, high-throughput, and inexpensive targeting of cellular biomarkers to address the heterogeneous nature of multiple diseases[43-48]. Surface marker profiling (e.g., EpCAM) of circulating tumor cells (CTCs) have shown clinical relevance with cancer progression[49,50]. We also found that cell subpopulations with different surface markers showed distinctive behaviors. For example, tumor-infiltrating lymphocytes (TILs) with medium CD39 expression are more potent in killing cancer cells compared with TILs with high or low CD39 expression[51]. We therefore hypothesized that biomarker-based profiling of sEVs could predict therapeutic outcomes.

Herein, we describe a microfluidic approach for the molecular profiling of sEVs through nanoscale exosomal protein-based sorting using immunomagnetic-activated cytometry (NanoEPIC). The NanoEPIC platform can conduct high-throughput and high-resolution sorting of individual sEVs based on surface marker expression. Furthermore, NanoEPIC performs single-step processing of sEVs directly from cell-free biological fluids such as plasma or cell culture medium. Using an mouse model, we validate the reliability of NanoEPIC for predicting immunotherapeutic responses and demonstrate the strong associations between exoPD-L1 profiling with tumor PD-L1 burden and T cell suppression. Additionally, sEVs can be efficiently recovered for downstream analysis, enabling the testing of suppression of CD8+ T cells. The activation of CD8+ T cells are inhibited differently after treatment with sEVs displaying varied levels of PD-L1. Overall, this approach not only allows for the phenotypic profiling of exoPD-L1 which is a potential strategy to monitor therapeutic responses but also enables the isolation of heterogeneous sEVs for further downstream investigations, offering capabilities for sEV-focused research.

## Results

### Development of the NanoEPIC platform

The NanoEPIC platform is designed to enable one-step sorting and molecular profiling of sEV that are pre-labeled with antibody-functionalized magnetic nanoparticles (MNPs) (Fig. 1). The working principle of the NanoEPIC system relies on the differential deflection of sEVs based on their varying biomarker expression (Fig. 1a). sEVs with higher expression of the target biomarker (e.g., PD-L1) bind to more MNPs and subsequently gain higher magnetic susceptibility. Separation of phenotypically distinct vesicles is facilitated through the directed movement of sEVs along ferromagnetic guides located below the microfluidic flow channels (Fig. 2a and Supplementary Fig. 1a). The magnetic guides were designed to branch out laterally from the inlet to outlet to generate a magnetic field gradient when magnetized by an external magnetic field, allowing for the separation of particles with varying magnetic susceptibility (Fig. 2b, c and Supplementary Fig. 1b–d). Made from the ultra-high permeability metal alloy Metglas 2714 A, the magnetic guides were precisely tuned to ensure an optimal balance between the Stokes' drag force ($F_d$), produced by the fluid flow, and the magnetic force ($F_m$), which acts on the MNP-labeled sEVs (see Supplementary Text). sEVs with higher MNP loading typically experience more lateral deflection toward the edges of the device. Based on the final lateral displacements, it is possible to collect sEVs bearing four different expression levels of the target biomarker: negative, low (exo-L), medium (exo-M), and high (exo-H). To establish a uniform laminar flow and minimize non-specific deflections, samples were flow-focused with an additional buffer stream (Supplementary Fig. 2a). Equipped with a customized 3D-printed magnetic stage, the NanoEPIC system allows for the simultaneous processing of 6 samples (Supplementary Fig. 2b). As this approach permits marker-based profiling of sEVs, the NanoEPIC can be easily applied for the screening of exoPD-L1 to predict immunotherapeutic outcomes (Fig. 1b). Additionally, the ability of NanoEPIC to perform high-resolution separation of individual sEVs provides opportunities for downstream analyses of specific sEV subpopulations (Fig. 1c). To optimize the performance of the NanoEPIC platform for exoPD-L1 sorting, we evaluated three cell lines with distinct levels of PD-L1 expression: PC9, H460, and H1975. Through flow cytometry and PD-L1 ELISA, we confirmed both the cPD-L1 and exoPD-L1 levels to be lowest in PC9 and highest in H1975 (Supplementary Fig. 3a, b). The magnetic loading on sEVs from different cell lines was inspected through transmission electron microscopy (TEM) and was found proportional to exoPD-L1 expression (Fig. 2d and Supplementary Fig. 4). Since H1975 sEVs displayed the broadest magnetic loading, we used H1975 as a model for optimizing the NanoEPIC platform for exoPD-L1 profiling. To alleviate the interference of larger EVs such as microvesicles (MVs) and apoptotic bodies (APOs) with larger sizes, samples were centrifuged at 10,000 g and then filtered with 0.22 μm filters during sample preparation. Therefore, while processing samples in NanoEPIC, only sEVs were introduced to the device. Also, larger vesicles require more PD-L1 to achieve magnetic deflection as the Stoke's force is bigger. Therefore, the NanoEPIC is optimal for sEV profiling.

There are two main components in the NanoEPIC device that are crucial for its performance: (1) the deflection angle of magnetic guides, and (2) flow channel height. The deflection angles allow for tuning the equilibrium between the fluid drag force and magnetic force which consequently dictates the degree of separation between the sorted sEV subpopulations. To determine the angular thresholds of our magnetic guides, we modeled 4, 10, and 20 as the approximate number of MNPs conjugated to sEVs in the low, medium, and high outlets respectively (Supplementary Fig. 5a). Through experimentation, we observed near-saturation deflection efficiency at a minimum deflection angle of 3° (Fig. 2e) which represents an acceptable starting point for exoPD-L1 segregation. To sort sEVs with distinct phenotypic profiles, we selected 3° for exo-L, 5° for exo-M, and 10° for exo-H subpopulations. As for the channel height, increased height allows for higher throughput but can

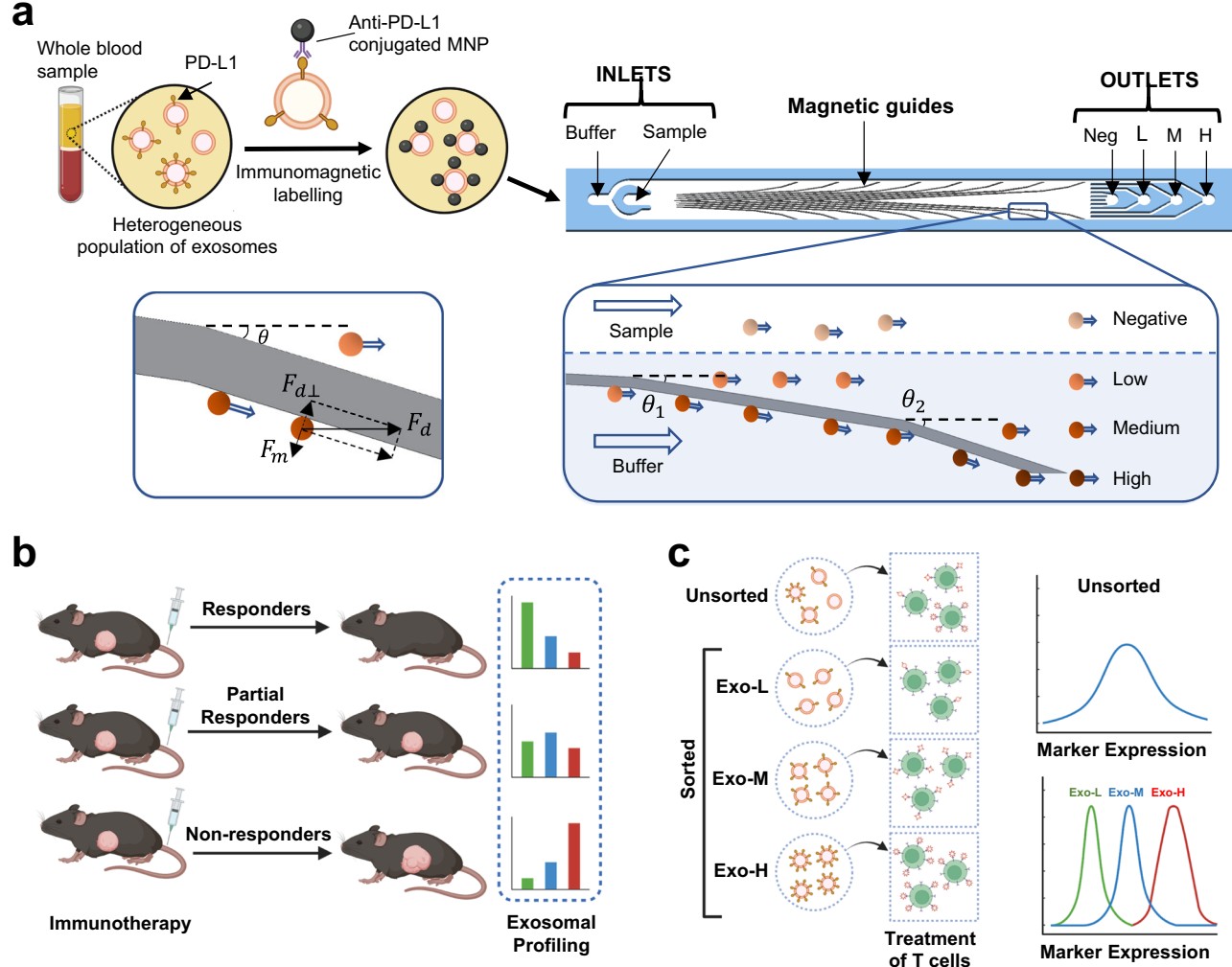

**Fig. 1 | Overview of the NanoEPIC Platform.** The NanoEPIC system relies on magnetic deflection to achieve phenotypic profiling and nanoscale sorting of sEVs. **a** Plasma extracted from whole blood was first treated with anti-PD-L1 conjugated MNPs. The MNPs-treated plasma sample is directly processed through NanoEPIC for exoPD-L1 profiling. The NanoEPIC device consists of a sample inlet and a buffer inlet. The sample is focused to the middle of the microchannel by the buffer during processing. Magnetic guides are embedded under the microchannel for exosomal sorting. Magnetically labeled sEVs experience a strong magnetic force in the direction normal to the magnetic guides due to the intensified magnetic gradient found at the edges of the guides. If the magnetic force is large enough to balance the Stokes' drag force in the direction normal to the magnetic guide, the magnetically labeled sEVs will divert from the initial flow path and follow the magnetic guide. Unlabeled particles are focused on the negative outlet. Deflected sEVs are collected in the low/medium/high outlets based on PD-L1 expression. **b** ExoPD-L1 is used as a predictive marker for cancer immunotherapeutic outcomes through exoPD-L1 profiling. To showcase the utility of the platform for monitoring the therapeutic response to immunotherapy, mouse models under anti-PD-1 therapy were tracked for exoPD-L1 profiling using the NanoEPIC platform. The exoPD-L1 profiles were assessed in light of treatment outcomes. **c** Sorted sEVs can be used for downstream analysis. As illustrated, sEVs from different outlets were used to interact with cytotoxic T cells. Treatment with different sEV subpopulations resulted in variations in the level of T cell inhibition.

also diminish the magnetic influence of the magnetic guides. Given that the magnetic guides are positioned below the device, sEVs migrate toward the bottom of the device to fully experience their designed magnetic trajectories. Both simulations and experimental outcomes suggest 30 μm to be the optimal height for exoPD-L1 sorting (Fig. 2f and Supplementary Fig. 5b–f). It is worth noting that an overly strong magnetic gradient might result in trapping the MNP-labeled sEVs at the bottom of the microchannel. This problem was addressed by adjusting the distance between the bottom of the channel and the magnetic guides (Supplementary Fig. 6a). Next, we optimized the sample flow rate to maximize the capture efficiency. Instead of anti-PD-L1, we used anti-CD9 as a sorting marker to determine the maximal sorting efficiency of sEVs. We demonstrated that a flow rate of 200 μL/h has led to the highest deflection efficiency (86.6 ± 2.9%) (Fig. 2g). To gauge the specificity of the NanoEPIC device, we generated mutagenic PD-L1 knockouts of H1975 (PD-L1 KO) (Supplementary Fig. 6b) and compared

the exoPD-L1 profile of PD-L1 KO and wildtype H1975 (WT). As depicted in Fig. 2h, the NanoEPIC platform maintained high specificity as indicated by the low deflection efficiency of PD-L1 KO (5.4 ± 1.9%) and isotype antibody control (8.6 ± 3.4%) compared to 64.2 ± 2.2% obtained using WT sEVs. Additionally, we found that most WT sEVs were collected in the exo-H outlet, whereas most of the PD-L1 KO sEVs or sEVs treated with isotype antibody-linked MNPs were collected in the exo-L outlet (Fig. 2i), which suggests that unlabeled sEVs might have escaped to the exo-L outlet with minimal nonspecific deflections into the exo-M and exo-H outlets.

## ExoPD-L1 profiling of tumor-derived sEVs with NanoEPIC
Having established the high sensitivity and specificity of NanoEPIC for sorting sEVs, we further assessed the performance of exoPD-L1 profiling using the NanoEPIC platform. First, we assessed exoPD-L1 profiling in H1975 sEVs at different flow rates. While flow rates of 200 μL/h

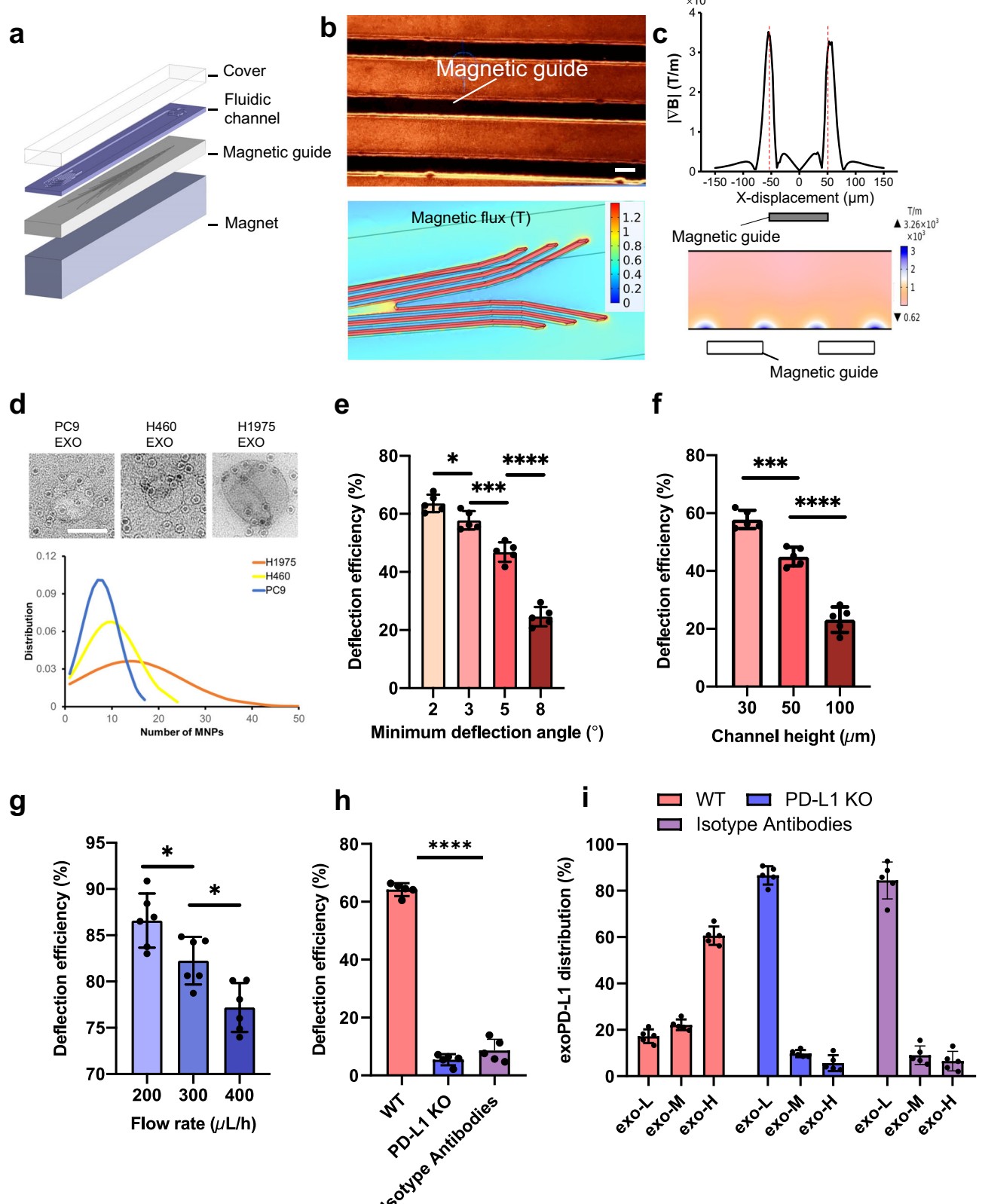

and 300 μL/h displayed nonsignificant differences in deflection efficiency (Fig. 3a), the distributions of sEVs between the deflected outlets were much more pronounced (Fig. 3b). More importantly, increasing the flow rate not only decreased the deflection efficiency but also shifted the distribution of sEVs towards the low-expression outlet which corroborated our simulations (Supplementary Fig. 7). Next, we utilized NanoEPIC for profiling exoPD-L1 in other cancer cell lines. The

deflection efficiency and exoPD-L1 profiles were significantly different among all three cell lines which agreed with our previous findings (Supplementary Fig. 8a, b). Since the expression of exoPD-L1 has previously been shown to be upregulated after treatment with IFN-γ[6,52], we also compared the exoPD-L1 profiles between IFN-γ treated and untreated cells (Supplementary Fig. 8c). Notably, the differences in exoPD-L1 profiles between IFN-γ treated and untreated cell lines were

**Fig. 2 | Design of the NanoEPIC device. a** NanoEPIC includes a glass substrate with magnetic guides patterned under a fluidic channel and a PDMS cover. A permanent magnet is positioned below the NanoEPIC to facilitate magnetic deflection. **b** Visualization of magnetic guides through (top) (scale bar = 100 μm) and the corresponding simulation of the magnetization of magnetic guides under an external magnetic field (bottom). **c** Simulated magnetic gradient across the magnetic guide. **d** The number of anti-PD-L1 conjugated MNPs bound to sEVs from different cell lines. (upper) Representative figures of sEVs bound to anti-PD-L1 conjugated MNPs obtained by TEM. Scale bar is 100 nm. In general, the number of MNPs bound to sEVs is proportional to exoPD-L1 expression. The average number of MNPs bound to sEVs collected from H1975 cells is the highest among the three tested cell lines. Furthermore, H1975 sEVs displayed the broadest distribution of the number of MNPs bound to each sEV. **e** Deflection efficiency of anti-PD-L1 MNP bonded sEVs in NanoEPIC with different minimum deflection angles. Smaller

deflection angles led to higher deflection efficiency. ($p = 0.0479$; $p = 0.0003$; $p < 0.0001$). **f** Deflection efficiency of anti-PD-L1 MNP bonded sEVs in NanoEPIC with different channel heights. Smaller channel height gives higher deflection efficiency. ($p = 0.0003$; $p < 0.0001$). **g** Deflection efficiency of sEVs bound with anti-CD9 MNPs. ($p = 0.0363$; $p = 0.0150$) **h** Specificity of the NanoEPIC assessed through the difference in deflection efficiency between H1975 wildtype (WT) sEVs and H1975 PD-L1 knockout (KO) sEVs, which were sorted based on exoPD-L1. Isotype antibodies were used to further confirm the specificity of magnetic labeling of sEVs based on exoPD-L1. ($p < 0.0001$) **i.** ExoPD-L1 profile distribution in sEVs from H1975 WT cells and PD-L1 KO H1975 cells. ExoPD-L1 profile of sEVs treated with isotype antibody-MNPs is also given. $n = 5$ biologically independent samples for **e-i**. All bar plots represent mean ± s.d. *$P < 0.05$, **$P < 0.01$, ***$P < 0.001$, ****$P < 0.0001$, one-sided unpaired t-test. Source data are provided as a Source Data file.

also detectable using the NanoEPIC platform (Fig. 3c–f), which highlights the high-resolution profiling efficiency of the NanoEPIC platform.

We next determined if the sorted sEVs could be recovered for subsequent downstream analysis. TEM imaging confirmed the number of MNPs bound to sEVs is correlated to their designated outlets (Fig. 3g); however, to enable the utility of sorted sEVs, the MNPs must be stripped off. Unlike cells, sEVs lack innate processes that can degrade, internalize, or detach MNPs. Using antibody elution buffer, we were able to remove most MNPs while maintaining the integrity of PD-L1 proteins on sEVs (Supplementary Fig. 9a-c). MNP release was validated through exoPD-L1 profiling using NanoEPIC (Fig. 3h, i). The eluted MNPs can be separated from sEVs through sucrose-gradient ultracentrifugation (Supplementary Fig. 9d, e). Using the purified sEVs, we further confirmed the difference in PD-L1 expression in sEVs collected from each of the outlets using PD-L1 ELISA and western blotting (Fig. 3j, k).

### In vivo analysis of exoPD-L1 for PD-1 immunotherapy

To examine the potential clinical utility of profiling exoPD-L1, we established a PD-1 immunotherapy mouse model through the injection of MC38 cells into C57BL/6 mice followed by a semiweekly administration of anti-PD-1 antibodies (Fig. 4a). We assessed whether exoPD-L1 profiling can be used to differentiate between the heterogeneous responses to anti-PD-1 immunotherapy (Supplementary Fig. 10). At the endpoint of the study (23 days post-inoculation), terminal blood collection was performed and the plasma was separated. The plasma was then used for exosomal analysis using the NanoEPIC platform. The status of complete responders, partial responders, and non-responders were determined through tumor volumetric measurements at day 23. Figure 4b represents a summary of the exoPD-L1 profiles from anti-PD1 treated mice (Supplementary Fig. 11a). The NanoEPIC analysis revealed a lower proportion of exo-L and a considerably higher proportion of exo-H sEVs found in non-responders compared to responders (Supplementary Fig. 11b) which suggests that elevated exoPD-L1 expression led to greater immunotherapeutic resistance. It was also observed that without any immunotherapeutic pressure, exoPD-L1 profiles were unpredictable and inconclusive (Supplementary Fig. 11c, d). We then investigated whether plasma levels of sEVs are correlated with outcome. The concentration of sEVs from all the mice were measured using NTA. No significant differences were observed between complete responders, partial responders and non-responders (Supplementary Fig. 10d). To refine the quantitation of exoPD-L1 expression measured with the NanoEPIC platform, we generated a numerical protein expression index called the NanoEPIC score. This score reports the overall exoPD-L1 expression while incorporating the deflection efficiency and exoPD-L1 profiling from the NanoEPIC's high-throughput and high-resolution nanoscale sorting of heterogenous sEVs (see supplementary text for the

calculation of the NanoEPIC score). Using this approach, we observed a direct correlation between tumor volume and NanoEPIC score (Fig. 4c), indicating that the NanoEPIC platform could potentially be used to monitor and predict tumor growth.

Tumor cPD-L1 levels have traditionally been used for the indication of PD-L1-associated tumor immune evasion; however, the acquisition of PD-L1+ cells from solid tumor tissues is highly invasive. On the other hand, cPD-L1-based liquid biopsy which relies on the analysis of circulating tumor cells (CTCs) can be challenging due to the scarcity of CTCs in early cancer stages. Since sEVs are highly abundant in the plasma and simple to obtain, we were interested in determining whether the levels of plasma exoPD-L1 would reflect the levels of cPD-L1 in tumors. Using our previously established prismatic deflection cell-sorting device (PRISM)[45], we profiled dissociated tumor cells from our C57BL/6 mice based on cPD-L1 expression (Fig. 4d and Supplementary Fig. 12a). Based on our findings, the distribution of exoPD-L1 closely reflected the distribution of PD-L1+ cells (Supplementary Fig. 12b–d), suggesting that exoPD-L1 expression is positively correlated with cPD-L1 expression.

To further ascertain the relationship between exoPD-L1 and cPD-L1, we evaluated an MC38 colorectal cancer model. For this investigation, we first segregated MC38 cells using the PRISM device into three subpopulations of varying cPD-L1 expression, which were promptly validated through flow cytometry (Supplementary Fig. 12e). We then collected sEVs from each of the sorted MC38 subpopulations and profiled their exoPD-L1 using the NanoEPIC device (Fig. 4e–h). The total fraction of sEVs containing PD-L1 was found to increase with higher cPD-L1 expression in MC38 cells (Fig. 4e). Upon examining the exoPD-L1 profiles, a similar trend was seen with a larger proportion of exo-L found in low cPD-L1 expressing MC38 and a higher proportion of exo-H in high cPD-L1 (Fig. 4f–h). Overall, NanoEPIC is a compelling prognostic platform for PD-1 immunotherapy as exoPD-L1 is highly abundant, easily obtainable, and correlates with PD-L1 burden in tumors to a similar degree as traditional prognostic markers such as tumor cPD-L1.

We next explored the relationship between exoPD-L1 and their immune effectors, particularly with CD8+ tumor-infiltrating lymphocytes (TILs). Flow cytometric analysis of tumors from partial responder, non-responder, and control (no anti-PD-1 immunotherapy) mice revealed a uniform decrease in CD45 + CD8+ TILs with higher NanoEPIC score (Fig. 5a and Supplementary Fig. 13a). It was also found that TILs with higher NanoEPIC score had reduced proliferation, T cell activation/differentiation, cytokine secretion and T cell cytotoxicity as demonstrated by the decrease in expression of ki67, TCF7, CD69, CD137, IFN-γ, and granzyme B (GzB) (Fig. 5b–g and Supplementary Fig. 13b–g). These findings are consistent with previous investigations, indicating a decline in TIL activity is strongly associated with higher exoPD-L1 levels[6,7,23]. While we expectedly detected a reduction in PD-1 expression with higher exoPD-L1, we also observed a striking

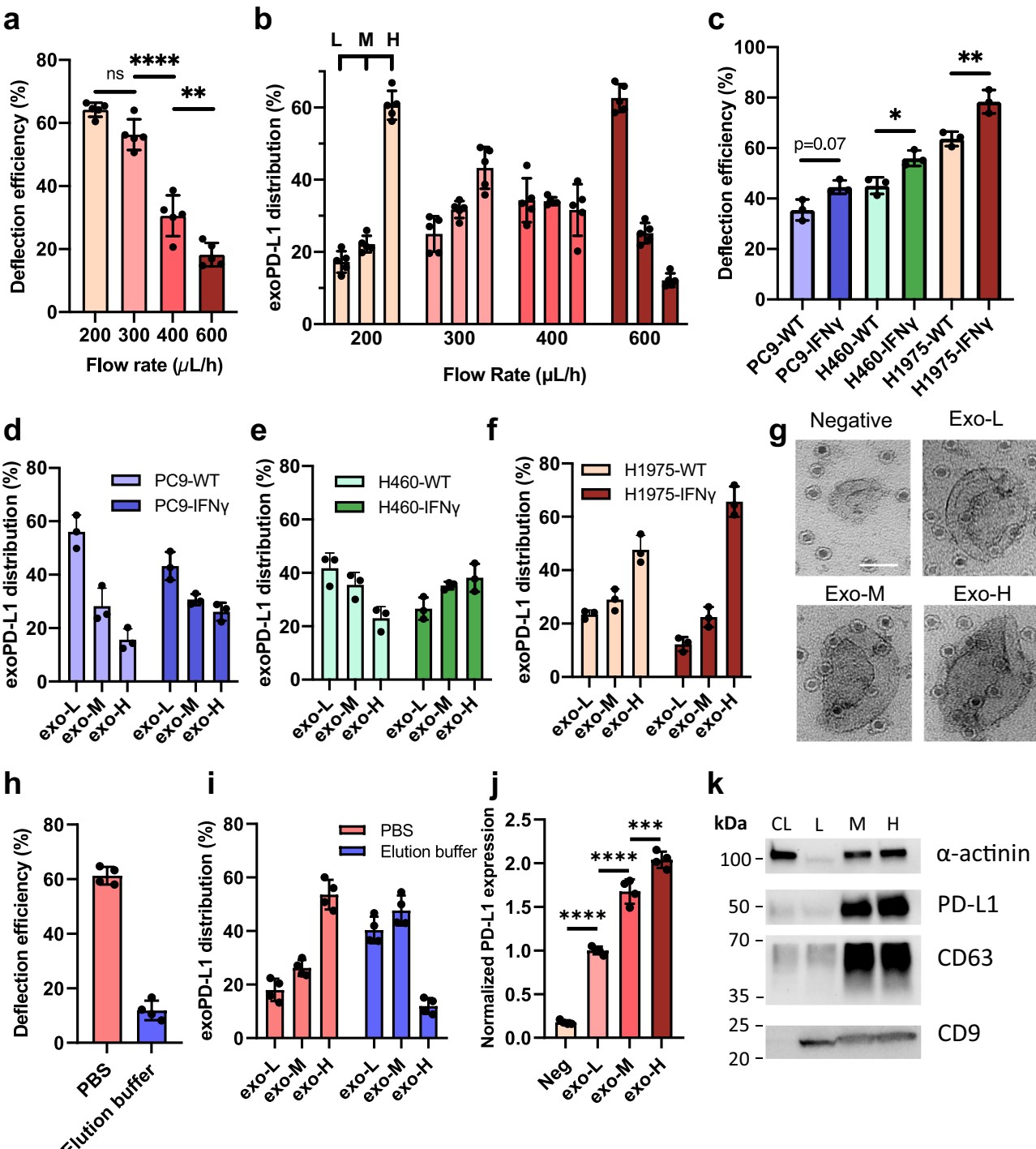

**Fig. 3 | Performance of the NanoEPIC platform. a** Deflection of anti-PD-L1 MNPs labeled sEVs from H1975 at different flow rates. ($p = 0.0651$; $p < 0.0001$; $p = 0.0031$). **b** ExoPD-L1 profiling of sEVs from H1975 at different flow rates. **c** Deflection efficiency of sEVs from different cell lines (H1975, H460, and PC9). The deflection efficiency of sEVs from corresponding cells treated by IFNγ is shown. ($p = 0.0713$; $p = 0.0250$; $p = .0029$). **d** ExoPD-L1 profiling of sEVs collected from WT PC9 cells and IFNγ treated PC9 cells. **e** ExoPD-L1 profiling of sEVs collected from WT H460 cells and IFN-γ treated H460 cells. **f** ExoPD-L1 profiling of sEVs collected from WT H1975 cells and IFN-γ treated H1975 cells. **g** TEM images of sEV subpopulations collected from Negative, Low, Medium, and High outlets of the NanoEPIC. The scale bar is 50 nm. **h** Deflection efficiency of anti-PDL1 MNPs bounded sEVs from H1975 cells after treatment with antibody elution buffer compared to control (i.e., sEVs treated with PBS). **i** ExoPD-L1 profiling of H1975 sEVs after treatment with elution buffer. PBS-treated sEVs were mainly collected in the exo-H outlet, whereas elution buffer-treated sEVs were mainly collected in the exo-L and exo-M outlets. **j** exoPD-L1 expression of H1975 sEV subpopulations collected from low/medium/high outlets, measured by PD-L1 ELISA. ($p < 0.0001$; $p < 0.0001$; $p = 0.0004$). **k** Western blot (WB) analysis of different sEV subpopulations compared to cell lysate (CL). $n = 5$ biologically independent samples for **a**,**b**. $n = 3$ biologically independent samples for **c**–**f**. $n = 4$ biologically independent samples for **h**–**j**. All samples were harvested from H1975 cells. WB assay was independently repeated 3 times for **k**. All bar plots represent mean ± sd. *$P < 0.05$, **$P < 0.01$, ***$P < 0.001$, ****$P < 0.0001$, one-sided unpaired t-test. Source data are provided as a Source Data file.

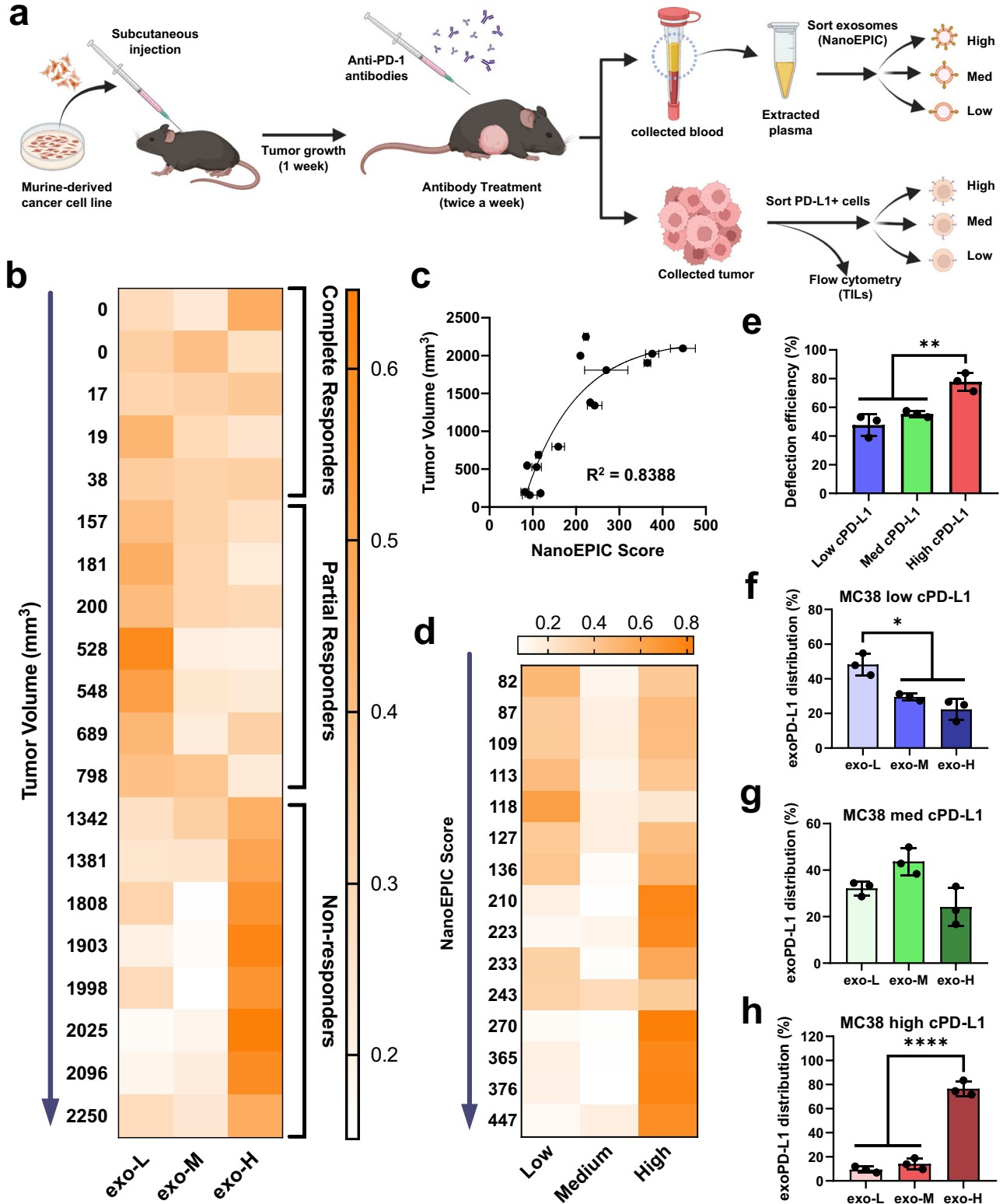

reduction in CD152 (CTLA-4) expression (Fig. 5h, i and Supplementary Fig. 13h, i). The CD152 pathway is involved in the inhibition of T cell activation which typically occurs in lymph nodes[53], suggesting that sEVs can elicit longer range immune-suppression beyond the tumor site. In summary, our results demonstrated the strong potential of using exoPD-L1 profiling for predicting T cell response to immunotherapy, further supporting NanoEPIC as a robust prognostic platform.

**Interaction of T cells and sorted sEV subpopulations**
Having demonstrated the varying extent of TIL inactivation in vivo relative to exoPD-L1 levels, we then explored whether NanoEPIC-sorted sEVs would exhibit distinct biological activities. The prevailing model for PD-L1-mediated immunosuppression relies on the interaction between PD-L1 with PD-1 generally expressed on CD8 cytotoxic T cells. To determine the direct link between exoPD-L1 and T cell suppression, we used an in vitro model in which isolated

**Fig. 4 | Analysis of exoPD-L1 in PD-1 immunotherapy mouse model. a** Workflow of sEV and tumor analysis in PD-1 immunotherapeutic mouse model. Briefly, mice injected with MC38 cells were treated with either anti-PD-1 antibodies or saline (control) twice a week. 23 days post-inoculation, mice were sacrificed for plasma and tumor collection. Collected plasma was subjected to exoPD-L1 analysis using NanoEPIC while solid tumors underwent PD-L1 sorting or TIL analysis through flow cytometry. **b** Distribution of sEVs sorted in the low, medium, and high outlets of the NanoEPIC chip are represented as a heat map and arranged in ascending order with respect to tumor volume. **c** Scatterplot of tumor volume against NanoEPIC score ($n = 3$). An exponential plateau model was used for the regression analysis which

resulted in the following fit: $y = 2213 - (2213 + 1945)e^{-0.008393x}$. **d** Distribution of tumor cells sorted through each outlet of the prismatic deflection chip with PD-L1 as the sorting marker. Data arranged in ascending order with respect to NanoEPIC score. sEVs from PD-L1 sorted MC38 cells were collected and processed through the NanoEPIC device and the deflection efficiency ($p = 0.0085$) **e** along with the exoPD-L1 profiles for the low ($p = 0.0106$) **f**, medium **g**, and high ($p < 0.0001$) **h** of cPD-L1 are shown. $n = 3$ biologically independent samples for **e**–**h**. All data represent mean ± s.d. *$P < 0.05$, **$P < 0.01$, ***$P < 0.001$, ****$P < 0.0001$, one-way ANOVA. Source data are provided as a Source Data file.

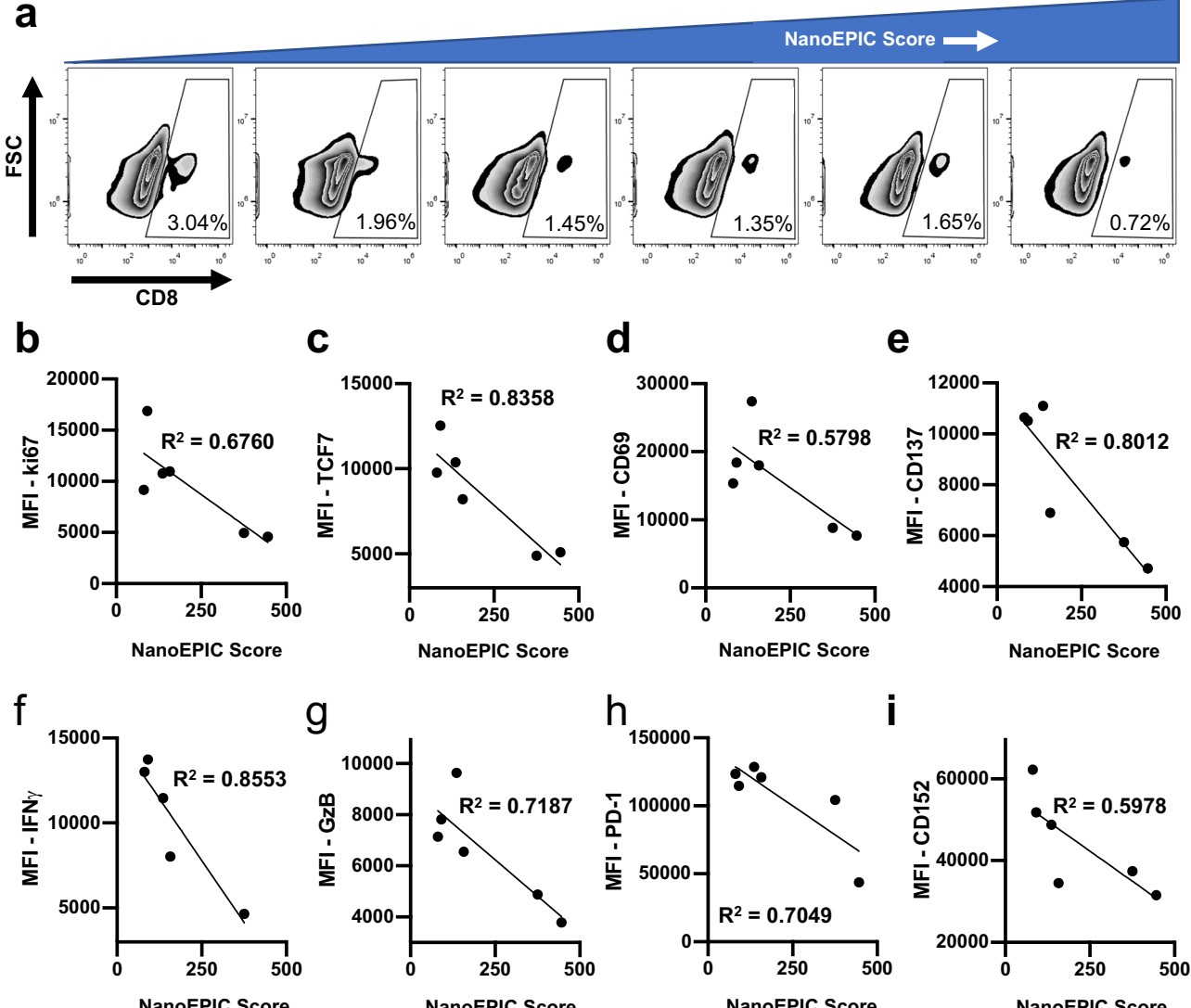

**Fig. 5 | Determination of NanoEPIC scores of TILs from murine PD-1 immunotherapeutic model. a** Proportion of CD45 + CD8 + T cells arranged with increasing NanoEPIC scores from left to right. Flow cytometric analysis of proliferation markers ki67 **b**, T cell differentiation and activation markers (TCF7, CD69,

and CD137) **c**–**e**, cytotoxicity (IFNγ, Granzyme B) **f**, **g** and immune-checkpoint markers (PD-1 and CD152) **h**, **i** shown as scatterplots of median fluorescence intensities and NanoEPIC score. All scatterplots were fitted with linear regression analysis. Source data are provided as a Source Data file.

CD8 + T cells were treated with PD-L1-sorted sEVs from the NanoEPIC device. First, we examined the relative binding of sEVs onto CD8 + T cells using SEM (Supplementary Fig. 14a). As seen in Fig. 6a, there were significantly more exo-H sEVs bound to CD8 T cells when compared with exo-L sEVs. To confirm this observation, we performed immunofluorescent staining of sEV-bound T cells using CD9 as an indicator of sEVs (Fig. 6b and Supplementary Fig. 14b).

Through measurement of CD9 fluorescent intensities, it was confirmed that there were statistically significant differences between the binding affinity of exo-L, exo-M, and exo-H sEVs onto T cells, with the lowest binding affinity exhibited by exo-L and the highest binding affinity in exo-H (Supplementary Fig. 14c). These results suggest exoPD-L1 levels are positively correlated to the binding affinity of sEVs to CD8 T cells.

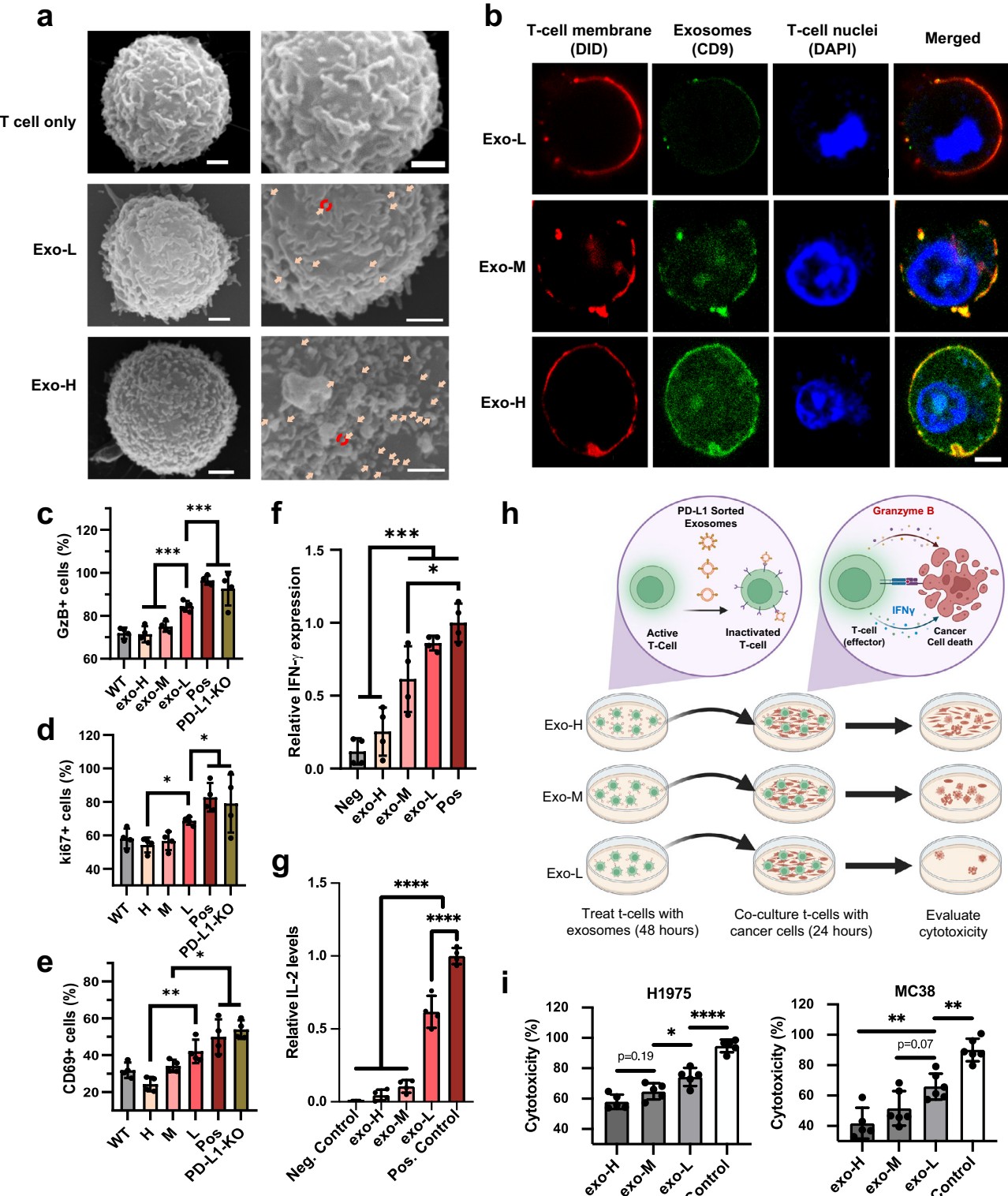

**Fig. 6 | Characterization of T cell properties after treatment with sorted PD-L1 + sEVs.** CD8 + T cells bound to different sEV subpopulations were imaged through SEM **a** and confocal microscopy **b**. Scale bar represents 1 μm for SEM and 4 μm for confocal microscopy. Proportion of T cells positive for granzyme B **c** ($p = 0.0030$; $p = 0.0058$), ki67 **d** ($p = 0.0101$; $p = 0.0350$), and CD69 **e** ($p = 0.0044$; $p = 0.0119$) was determined through flow cytometry after treatment with sEVs from exo-L, exo-M, exo-H, unsorted (WT), activated CD8 + T cells with no exosomal treatment (Pos), and exosomes acquired from PD-L1 knockouts (PD-L1-KO). Secretion of IFN-γ **f** ($p = 0.0002$; $p = 0.0140$) and IL-2 **g** ($p < 0.0001$; $p < 0.0001$) by

T cells after treatment of different sEV subpopulations was determined using ELISA. **h** Schematic of CD8[+] T cell cytotoxicity assay after treatment with varying exoPD-L1 subpopulations of sEVs. **i** T cell cytotoxicity assay from CD8[+] T cells cocultured with either H1975 (left) ($p = 0.1872$; $p = 0.0418$; $p < 0.0001$) or MC38 (right) ($p = 0.015$; $p = 0.0710$; $p = 0.0024$) after treatments from exo-L, exo-M, exo-H, or no sEVs (control). $n = 4$ biological independent samples for **c**–**g**. $n = 5$ for **i**. All bar plots represent mean ± s.d. *$P < 0.05$, **$P < 0.01$, ***$P < 0.001$, ****$P < 0.0001$, one-way ANOVA. Source data are provided as a Source Data file.

Next, we sought to investigate the phenotypic variations of T-cells after sEV treatment. Pre-activated CD8 + T cells were treated with sEVs from each outlet of the NanoEPIC device (exo-L, exo-M, exo-H) separately. Compared with the positive control (Pos) where pre-activated CD8 T cells were not treated with sEVs, a decrease in activation, proliferation, and cytotoxicity of T cells was observed through flow cytometry after treatment with exo-L, exo-M, and exo-H as indicated by the decreasing proportions of CD69, ki67, and GzB positive CD8 T-cells (Fig. 6c–e and Supplementary Fig. 15). Exo-H caused stronger inhibition of T cells compared with Exo-L. As comparison, treatment with unsorted sEVs (WT) was consistently found to be within the margins of exo-L to exo-H for all analyses. Through ELISA, it was shown that the cytokines, namely IFN-γ and IL-2 were similarly reduced in T cells when treated with sEVs with higher exoPD-L1 expression (Fig. 6f, g). For the cytokine secretion assay, fresh purified inactivated CD8 T cells were used as a negative control (Neg). To further evaluate the effects of exoPD-L1 on the cytotoxic activity of CD8 T cells, the cells treated with distinct sEV subpopulations were co-cultured with tumor cell lines (Fig. 6h). Through LDH assay, we observed a significant reduction in CD8 T cell cytotoxicity when treated with sEVs with higher exoPD-L1 levels (Fig. 6i). Overall, our findings demonstrate that the NanoEPIC is capable of sorting sEVs with biologically distinct activities and allow for the separation of unique sEV subpopulations.

### Validation of NanoEPIC platform with clinical specimens

To establish that the NanoEPIC platform could be used with clinical specimens, we analyzed a small set of samples from patients undergoing anti-PD-1/anti-PD-L1 immunotherapy. Plasma samples were gathered from 10 de-identified cancer patients (multiple visits) and 5 healthy donors to determine whether exoPD-L1 levels could be measured (Supplementary Table 4). Comparing patient samples to healthy donor samples, we observed a significant elevation in exoPD-L1 levels (Fig. 7a). We also analyzed changes in NanoEPIC scores before and after treatment. Notably, responders exhibited a consistent decrease in exoPD-L1 expressions (Fold change<1), while non-responders showed a general increase (Fig. 7b and Supplementary Fig. 17). Additionally, we observed a positive correlation between changes in NanoEPIC scores and tumor size (Fig. 7c). These findings indicate that

NanoEPIC could be applied in the future to predict immunotherapy responses and aid in treatment selection. Moreover, the ability of the NanoEPIC platform to distinguish between healthy and clinical samples highlights its potential diagnostic capabilities.

## Discussion

Here, we have presented a platform that utilizes magnetic-activated ranking for nanoscale cytometry of sEVs. We conceptually validated the feasibility of magnetic labeling of sEVs with MNPs and demonstrated nanoscale deflection of individual sEVs. The optimized NanoEPIC can perform phenotypic profiling with high efficiency, specificity, and throughput. As a microfluidics-based platform, the NanoEPIC is advantageous as it is simple to fabricate, low-cost, and user-friendly.

The NanoEPIC platform can be adapted to a variety of applications. Since the magnetic deflection of sEVs relies on the binding of antibody-decorated MNPs, NanoEPIC permits sorting sEVs based on the level of any surface marker. The NanoEPIC's capability to segregate sEVs based on surface marker expression allows for precise analysis of specific sEV subpopulations that were previously not feasible. While sEVs have been used to examine the performance of NanoEPIC, this nanoscale cytometry system can be widely adapted for the sorting and phenotypic profiling of other submicron/nanoscale particles such as certain bacteria or subcellular organelles. To our knowledge, few other technologies can perform high-resolution molecular sorting and profiling at that throughput.

Overall, we demonstrated how NanoEPIC could potentially be used as a cancer diagnostic tool. The NanoEPIC was used in mouse models to monitor the response to anti-PD1 immunotherapy and produce signature exoPD-L1 profiles that can be used to distinguish between responders and non-responders to immunotherapy. Our the numerical NanoEPIC scores not only reflected tumor progression but also demonstrated a high correlation with the suppression of TILs. The downstream analysis showed that different sEV subpopulations with varying PD-L1 expression exhibited different inhibitory behaviors on T cells. Given that different exosomal biomarkers such as CTLA4, TIM3, CD47 are being studied for the prediction of immunotherapy response, the NanoEPIC can be used to profile panels of exosomal

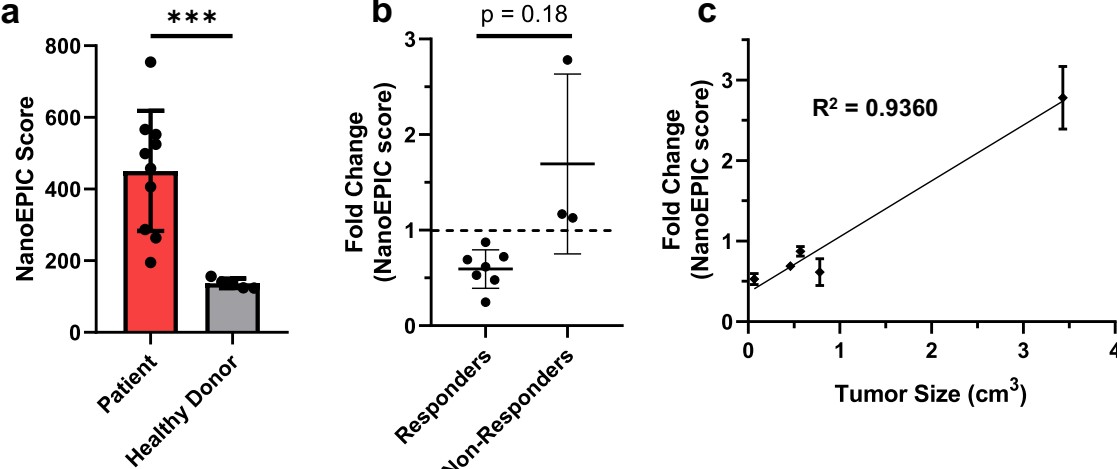

**Fig. 7 | Assessment of clinical specimens with the NanoEPIC Platform.** To evaluate the performance of the NanoEPIC platform on specimens collected from patients receiving anti PD-1/PD-L1-based immunotherapy, a total of 10 de-identified plasma samples were processed using the NanoEPIC device for exoPD-L1 profiling. **a** ExoPD-L1 profiling of patient samples before immunotherapy was compared with that of healthy donors. ($n = 10$) ($p = 0.0002$). **b** Changes in NanoEPIC scores were compared between responders and non-responders to immunotherapy. The fold change in NanoEPIC score represents the ratio of NanoEPIC score after immunotherapy over NanoEPIC score before immunotherapy. A fold change of 1 signifies no change in NanoEPIC score. The correlation between fold change in NanoEPIC score and **c** tumor size obtained through CT scans. $n = 3$ technical replicates for **a–c**. All bar plots represent mean ± s.d. ***$P < 0.001$, one-sided unpaired t-test. Source data are provided as a Source Data file.

biomarkers and ultimately be applied for further clinical investigations to guide clinical decisions and improve therapeutic outcomes. In addition to the diagnostic utility of NanoEPIC, therapeutic sEVs are emerging as promising drug delivery agents due to their excellent biocompatibility, low immunogenicity, persistence in the blood, and ability to cross various biological barriers[54]. Several reports have demonstrated the utility of sEVs as drug delivery vehicles for drug-resistant lung cancer[55–57]. Hence, the ability to isolate distinct sub-populations of sEVs with favorable phenotypes and superior delivery properties would offer many opportunities for the treatment of lung cancer[54]. This necessitates high-throughput methods capable of accurately sorting sEVs from large volumes of patients' blood, which suggests a future utility for the platform. Given that drug loading to sEVs can also be heterogeneous[58], the NanoEPIC system can be used to sort out sEV subpopulations with high drug loading, which can potentially reduce the dose of sEVs required for effective treatment. Another application of NanoEPIC can be sorting of cargo-loaded sEVs based on their specificity to recipient cells. sEVs engineered with antibodies specific to recipient cells have been used for cargo delivery[59]. Sorting of high marker expressed sEVs can increase the uptake of cargo-loaded sEVs by the recipient cells. Also, we have shown that the NanoEPIC system was able to sort magnetically labeled concentrated sEVs (up to $10^{12}$ particles/mL) with high deflection efficiency (more than 80%) at a flow rate of 200 μL/h. The NanoEPIC was amenable for simultaneous processing of up to 6 microfluidic chips, which further increases its total throughput.

While the NanoEPIC exhibits merits for biomarker profiling and sorting, as a magneto-activated system, it does lack the flexibility of simultaneously detecting multiple markers like FACS. More work must be done to multiplex biomarker profiling and sorting. One potential way to overcome this problem is through serial sorting with multiple magnetic labeling steps[59,60]. Another way is to use MNPs with distinctive magnetic properties such as size and magnetic susceptibility[61–63]. Continued efforts to assess novel magnetic sorting methodologies for facilitating multiplex analyses on the NanoEPIC platform hold tremendous promise in expanding the scope of extracellular vesicle investigations. Such advancements have the potential to uncover broader clinical implications on a larger scale.

## Methods
### Ethical statement
This research complies with all relevant ethical regulations and the related protocols were approved by the University of Toronto Research Ethics Board (REB). All animal experiments were operated following the protocols approved by the University of Toronto Animal Care Committee. Human plasma samples were obtained from PrecisionMed LLC upon the approval by the Western Institutional Review Board® (Puyallup, WA, USA) and after informed consents were signed. Whole blood samples were collected from 10 patients under anti-PD1/anti-PD-L1 immunotherapy ($n = 8$ female donors and $n = 2$ male donors aged 41–76) (Fig. 7, Supplementary Fig 17 and Supplementary Table 4). Plasma samples were extracted from whole blood immediately after blood collection and stored at −80 °C. In all cases, participants or legal representatives were informed in detail about the research purpose and informed consent was signed.

### Fabrication of NanoEPIC
The design of NanoEPIC was sketched using a CAD software (AutoCAD). Chrome masks containing the designed patterns were printed using a mask maker (Heidelberg μPG 501). The fabrication of NanoEPIC includes three main procedures: (i) patterning of magnetic guides, (ii) generation of microchannels, and (iii) binding of the devices (Supplementary Fig. 16). For patterning the magnetic guides, a sheet of Metglas 2714 A was glued onto borosilicate glass wafers (452, UniversityWafer) with a thin spun-coated layer of epoxy

(7370A38, McMaster-Carr) and left to dry overnight. Excessive epoxies were removed with acetone and isopropanol (IPA). The magnetic guides were photolithographically patterned onto a layer of 2 μm thick S1811 photoresist that was spun-coated onto Metglas. After developing the S1811 with MF-CD-26, Metglas was wet-etched using an etchant consisting of 3.6% HCl and 14.3% $H_2O_2$ in water. For the generation of microchannels, the S1811 photoresist was stripped with AZ300T (Integrated Micromaterials) and coated with Omni-coat (Kayaku Advanced Materials) to improve SU-8 photoresist adhesion. SU-8 3035 was used to develop a 40 μm-thick SU-8 layer to encapsulate the magnetic guides and a second 30 μm-thick SU-8 layer was developed on top to generate the microchannels. For device binding, PDMS cover pieces that were hole punched at the inlet and outlet sites were first treated with plasma for 1 min, then treated with 10% APTES in water for 30 min. The pieces were then rinsed with deionized water and bonded onto the SU-8 channels. The bonded chips were incubated at 70 °C overnight with vertical pressure to enhance the binding strength.

### Cell culture
Human NSCLC cell lines H1975 (CRL-5908, ATCC), H460 (HTB-177, ATCC), and PC9 (90071810, Sigma) were cultured in RPMI medium (350-007-CL, Wisent) while mouse colon cancer cell line MC38 (ENH204-FP, Kerafast) were cultured in DMEM (D5796, Sigma). All culture media were supplemented with 10% FBS (Wisent), 1% penicillin-streptomycin (Sigma), and incubated at 37 °C with 5% $CO_2$. For the stimulation of cells with IFN-γ, cells were incubated in medium supplemented with 100 ng/mL of either human or mouse IFN-γ (Peprotech) for 48 h before the collection of cell and sEV samples.

Human CD8 T cells were isolated from human blood mononuclear cells (70025, Stemcell Technologies) using Human CD8 + T cell isolation kit (130-096-495, Miltenyi Biotec) while mice T cells were collected from the spleen of C57BL/6 mice (Jackson Laboratory) and isolated with mouse CD8 + T cell isolation kit (130-096-543, Miltenyi Biotec). Human T cells were cultured in T cell expansion medium (10981, STEMCELL Technologies) with 10 ng/mL of human IL-2 (78036.3, STEMCELL Technologies). Mouse T cells were cultured in RPMI medium supplemented with 10% FBS and 10 ng/mL of mouse IL-2 (130-120-334, Miltenyi Biotec). T cells were activated with Human CD3/CD28/CD2 activation kit (10990, STEMCELL Technologies) at 25 μL/mL or Mouse CD3/CD28 beads (11456D, Thermofisher) at 25 μL/mL at every second passage.

### Genetic modifications
To generate the PD-L1 knockout (KO) H1975 cell lines, H1975 wildtype (WT) cells were first infected with lenti-Cas9 (52962LV, Addgene) followed by blasticidin selection to establish Cas9 expressing cell lines (H1975-cas9). The H1975-cas9 cells were then simultaneously infected with two different guide RNA targeting PD-L1 (target sequence: GGTTCCCAAGGACCTATATG and CGCTGCATGATCAGCTATGG) that were previously packaged into lentivirus by Thermofisher (A32042). Monoclonal expansion was performed following puromycin selection and the knockout clones were identified with flow cytometry (cytoFLEX S, Beckman Coulter) for surface PD-L1.

For the modification of MC38 cell lines, gRNA used to knockout PD-L1 (target sequences: GTTTACTATCACGGCTCCAA and GGGGA-GAGCCTCGCTGCCAA) were purchased from Integrated DNA Technologies and lentiviral plasmids were obtained from Genscript (pLentiCRISPRv2 backbone) respectively. Lentiviral packaging was performed through co-transfection of HEK293T cells with psPAX2, pMD2.G, and the transfer plasmid. The medium was changed after 24 h and then collected at 24 h intervals. MC38 cells were infected with the lentivirus and selected with 10 μg/mL of puromycin. Fluorescent-activated cell sorting (BD FACSAria IIIu cell sorter) was performed to select cells negative for PD-L1 from the knockout population. Flow

cytometry was used to confirm the stable maintenance of PD-L1 expression levels.

## Purification of sEVs

For sEVs collected from cultured cells, the base culture medium was replaced with medium supplemented with 10% sEV-depleted FBS (Gibco) when cells reached ~70% confluency. Cells were then cultured for 48 hours then the culture medium was collected for sEV isolation. The culture medium was centrifuged at 2000 g for 30 minutes to remove dead cells and cellular debris followed by another centrifugation at 10,000 g for 1 h to remove large microvesicles. The supernatant were filtered with 0.22 μm filter and then mixed with an sEV precipitation kit (4478359, Thermofisher) at a 2:1 volume ratio, incubated overnight, and then centrifuged at 10,000 g for 1 h to pellet the sEVs. SEVs were resuspended with 1% bovine serum albumin (BSA, Sigma) with protease inhibitor (11836170001, Roche) and phosphatase inhibitor (4906845001, Roche) in PBS. All buffers were filtered with 0.22 μm filters (SLGV033R, Sigma) prior to mixing with sEVs. For blood samples, plasma was extracted by collecting the supernatant after 2-step centrifugation at 2000 g for 20 minutes followed by 10,000 g for 20 min. Samples were directly processed through the NanoEPIC device as discussed later.

## Magnetic labeling of sEVs

10 nm diameter protein-G cross-linked iron oxide magnetic nanoparticles at 1 mg/mL (IPG-10-01) were purchased from Ocean NanoTech and mixed with either human anti-CD9 antibody (10626D, Thermofisher), human anti-PD-L1 antibody (14-5983-82, Thermofisher), or mouse anti-PD-L1 (14-5982-82, Thermofisher) in a 15:1 volume ratio and chilled overnight at 4 °C. The antibody-bead conjugates were then pelleted through centrifugation at 100,000 g for 45 minutes and then resuspended in 0.1% BSA to reach a final concentration of 0.5 mg/mL and stored at 4 °C until later use. For the labeling of sEVs, $10^9$ sEVs in 100 μL of 1% BSA were mixed with 5 μL of the functionalized magnetic nanoparticles (MNPs) and agitated overnight at 4 °C prior to processing through the NanoEPIC device. To ensure the maximum interaction between EVs and MNPs and maximized magnetic labeling efficiency, an excessive amount of functionalized MNPs were added to the sEV sample. For the preparation of sEV-spiked plasma samples, 100 μL of sEVs in 1% BSA were mixed with the same volume of plasma and treated with 10 μL of anti-PD-L1 conjugated MNPs at 4 °C overnight. As a control, 100 μL of 1% BSA was mixed with the same volume of plasma and treated with 10 μL of anti-PD-L1 conjugated MNPs at 4 °C overnight. For mouse plasma samples, 100 μL of 2-fold diluted plasma was treated with 5 μL of mouse anti-PD-L1 conjugated MNPs at 4 °C overnight.

## Processing sEV samples through NanoEPIC

Before chip processing, all buffers were filtered through a 0.22 μm syringe filter and degassed under a vacuum for 15 minutes. NanoEPIC chips were first treated with 0.1% Pluronic F-68 (Thermofisher) in deionized water for 15 min then washed with running buffer (1% BSA in PBS) to minimize non-specific adhesion. The device was then positioned on 4 laterally stacked N52-Neodymium magnets (BY042-N52, K&J Magnetics) for all further processing. 100 μL of magnetically labeled sEV samples were mixed in a 1:1 volume ratio with running buffer and loaded into the NanoEPIC device. Fluidic processing was performed through a syringe pump (Fusion 100, Chemyx) that was set to withdraw at a flow rate of 200 μL/h. Subsequent to sample processing, the devices were washed with PBS at a flow rate of 400 μL/h.

## Characterization of sEVs

The size distribution and concentration of sEVs were determined through nanoparticle tracking analysis (NTA) with NanoSight NS300 (Malvern). Transmission electron microscopy (TEM) was performed

for the visualization of sEVs binding to beads. Samples for TEM were fixed at a 1:1 volume ratio of 4% paraformaldehyde (PFA, F8775-4X25ML, Sigma) diluted in PBS. Specimen preparations were then performed by the Nanoscale Biomedical Imaging Facility (NBIF) in Peter Gilgan Centre for Research and Learning at The Hospital for Sick Children (Toronto, CA). Briefly, the sample was dropped on a glow-discharged formvar/carbon-supported copper grid. After staining with 2% uranyl acetate, the grids were air-dried and visualized using a Hitachi HT7800 transmission electron microscope.

## Separation of sEVs from antibody-functionalized MNPs

Two elution buffers were tested, including glycine-HCl (Sigma) and Pierce IgG elution buffer (Thermofisher). For glycine-HCl, 200 mM and 300 mM were tested. SEVs collected from the outlets of the NanoEPIC device were treated with an equal volume of elution buffer and immediately incubated at 37 °C for 30 min. The sEV samples were then separated from eluted MNPs through sucrose gradient centrifugation (0.5 M and 1.5 M sucrose concentration) running at 120,000 g for 1.5 h. sEVs were collected from the interface between the 0.5 M and 1.5 M sucrose while the released MNPs were pelleted at the bottom of the tube. To remove the excessive sucrose, the collected sEVs were washed with 5 mL of PBS and pelleted through centrifugation at 100,000 g for 2 h.

## Western blot analysis

Cell lysates were prepared by adding RIPA buffer (PI89900, Thermofisher) supplemented with protease inhibitor and phosphatase inhibitor to cell samples followed by a 14,000 g centrifugation for 15 minutes to collect the supernatant. sEV lysates were prepared by adding an equal volume of RIPA buffer and incubated at room temperature for 30 min. Lysates of different sEV subpopulations were normalized based on their total protein weight measured by BCA protein assay (Thermofisher). Lysates were separated using a 4–15% SDS-PAGE and transferred onto a nitrocellulose membrane. Blots were blocked with 3% BSA in Tris-buffered saline with Tween 20 (TBST) and incubated with primary antibodies at 4 °C overnight at the recommended supplier concentrations. For secondary staining, HRP-conjugated antibodies were added and left at room temperature for 1 h. The blots were developed with ECL substrate (Pierce). Information on the primary and secondary antibodies can be found in Supplementary Table 2.

## ELISA

All lysate samples were prepared as described above. The quantifications of protein through ELISA were performed by following the protocols from the supplier. Information on the ELISA kits can be found in Supplementary Table 3.

## Flow cytometry

Flow cytometric analysis was used to determine the protein expression in cells. Samples were typically fixed with 4% PFA followed by 20 minutes of blocking in 2% BSA. After primary antibody staining, samples were then washed and processed through cytoFLEX S flow cytometer (Beckman Coulter). For intracellular staining, samples were permeabilized with 0.2% Triton-X-100 (Sigma) after fixation. Information on the antibodies can be found in Supplementary Table 2. The collected data were analyzed using FlowJo.

## Immunotherapeutic mouse model

All animal experiments were operated in accordance with the protocols approved by the University of Toronto Animal Care Committee, with no tumors above the maximal limit of 2000 mm³ and tumor size did not exceed 20 mm in any directions. For our syngeneic mouse model, female C57BL/6 mice (6–8 weeks old) purchased from Jackson Laboratory were subcutaneously injected with $6 \times 10^6$ MC38 cells in

100 μL of PBS. Starting one week after inoculation, 200 μg of anti-PD-1 antibodies (ICH1132UL, Ichor. Bio) were administered intraperitoneally twice a week (control mice were injected with saline). Tumors were measured twice a week with a digital caliper and volume was calculated using the following formula: (width)$^2$ × length/2. Mice were euthanized when the tumors exceeded 20 mm in one dimension or 23 days after inoculation. Blood was immediately collected through cardiac puncture and tumors and spleen were surgically removed. Plasma was isolated from blood samples for exosomal sorting through the NanoEPIC device. For mouse CD8+ T cell isolation, spleens were homogenized with a cell scraper (83.3950, Sarstedt), filtered through a 70 μm mesh cell strainer (352350, Falcon), and treated with RBC lysis buffer (00-4333-57, Thermofisher) before magnetic separation. Tumor cells were dissociated with Tumor Dissociation Kit (130-096-730) from Miltenyi Biotec and were either analyzed for tumor-infiltrating lymphocytes (TILs) through flow cytometry or sorted for PD-L1 through our previously reported prismatic deflection chip (PRISM). For data analysis, mice were categorized into 3 groups based on their differential responses to immunotherapy: "complete responders" referred to mice that had a significant reduction in tumor volume, "partial responders" displayed no significant changes in tumor volume, and "non-responders" displayed a significant development in tumor volume from the initiation of immunotherapy to the endpoint of the study (23 days post inoculation) (Supplementary Fig. 10a, b).

### T cell-sEV binding assay

To facilitate the binding of sEVs to CD8 T cells, 100 μL of CD8+ T cells in fresh medium were dispensed into 96-well plates at a concentration of 2 × 10$^6$ cells/mL. Cells were activated for 24 h with the addition of CD3/CD28/CD2 for human lines or CD3/CD28 beads for mice lines as mentioned earlier. Inactivated CD8+ T cells used as controls were supplemented with PBS instead of the activation cocktail. Preactivated CD8+ T cells were treated with 10 μL of sEV subpopulations (1 mg/mL) for 48 h at 37 °C and 5% CO$_2$. After sEV treatment, CD8+ T cells under different conditions were harvested and used for flow cytometry to measure T cell activation, proliferation, and cytotoxicity. T cell culture media were collected to measure cytokine secretion with ELISA.

### Visualization of sEV-bound T cells

To visualize the binding of sEVs to T cells, scanning electron microscopy (SEM) was used. 100 μL of preactivated CD8+ T cells (10$^6$ cells/mL) were treated with 10 μL of sEVs (1 mg/mL) for 2 h at 37 °C. For SEM, we first immobilized the T cells onto gold electrodes functionalized with anti-CD8. Primary fixation was achieved through overnight incubation of samples in 0.1 M phosphate buffer containing 4% paraformaldehyde and 1% glutaraldehyde. Samples were then additionally fixed with 1% osmium tetroxide in 0.1 M phosphate buffer for 30 minutes. Samples were dehydrated using a gradually increasing ethanol series, followed by critical point drying. Dehydrated samples were then spin-coated in gold and imaged using the Prisma E scanning electron microscope.

Confocal microscopy was additionally used to verify the relative binding of different sEV subpopulations to T cells. The T cell-sEV binding protocol is modified to improve the fluorescence intensity. 100 μL of sEVs (1 mg/mL) of each subpopulation were incubated with 5 μL of Anti-CD9-PE (Thermofisher) overnight at 4 °C. SEV samples were then washed with PBS at 100,000 g for 2 h. The pelleted sEVs were then resuspended in 100 μL of PBS. For cell staining, 100 μL of activated T cells (10$^6$ cells/mL) were incubated with 1 μL of DiD lipid dye (V22887, Thermofisher) for 15 minutes at 37 °C. Cells were washed 4 times with PBS and then blocked in 400 μL of 1% BSA for 15 min. For each condition, 10 μL of PE-stained sEVs were added to the 100 μL of T cell samples and incubated at 37 °C for 2 hours. Samples were then washed with 1 mL of PBS to remove unbound sEVs, then fixed with 4%

PFA for 15 minutes. After another PBS wash, 1 drop of nuclear stain (R37606, Thermofisher) in 100 μL of PBS was added and samples were incubated for 10 minutes at room temperature. After a final wash with PBS, samples were then imaged using the Zeiss LSM 880 microscope.

### Cytotoxic CD8 T cell-mediated tumor killing assay

To measure the effect of exoPD-L1 on T cell-induced tumor cytotoxicity, we co-cultured sEV-treated CD8+ T cells with tumor cells and performed LDH-cytotoxicity assay (ab197004, Abcam). Briefly, sEVs from H1975 WT and MC38 WT were sorted through the NanoEPIC device and subsequently used to treat T cells using the aforementioned protocols. PD-L1 KO H1975 and PD-L1 KO MC38 cells were seeded into 96-well plates at 5,000 cells/well and incubated at 37 °C for 24 h. For co-culture, 100 μL of sEV-treated T cells (1.5 × 10$^5$ cells/mL) were added to the tumor cells and cultured for 48 h. LDH assay was subsequently performed following the supplier's protocol (Abcam).

### Clinical sample processing

Plasma samples were obtained from de-identified cancer patients who were undergoing anti-PD-1 or anti-PD-L1 immunotherapy, procured from PrecisionMed LLC (Supplementary Table 4). These samples were voluntarily collected by PrecisionMed LLC after approval from Western Institutional Review Board® (WIRB®) (Puyallup, WA, USA) (Protocol No.: 6054 WIRB® Protocol #20161289). Samples were obtained from $n = 8$ females and $n = 2$ males aged 47–76. All plasma samples were pre-filtered with 0.22 μm filters. The concentration of sEV in the plasma was measured to be between 0.6 and 1.74 × 10$^{11}$ particles/mL. For sample processing, 50 μL of plasma sample was first diluted 4 times with 1% filtered BSA. The diluted sample was then treated with 2.5 μL of anti-PD-L1 conjugated MNPs and incubated at 4 °C overnight. The mixture was then processed through the NanoEPIC system at 200 μL/h. sEVs collected from different outlets were analyzed with NTA as mentioned above. Gender analysis was not performed due to the limited number of available samples.

### Statistical analysis and reproducibility

All statistical analyses were performed with GraphPad Prism version 9.0.0. All graphical data are represented as mean ± s.d. unless stated otherwise. Each dot represents a biological replicate unless specified otherwise. Unpaired student t-test was used to compare two different groups while two-way ANOVA was used to compare multiple groups. $P$ value < 0.05 was considered statistically significant. No statistical method was used to predetermine sample size. No data were excluded from the data analyses. The investigators were blinded to allocation of treatment during mice experiments but not blinded when assessing outcome.

### Reporting summary

Further information on research design is available in the Nature Portfolio Reporting Summary linked to this article.

## Data availability

The main data supporting the results in this study are available within the paper and its Supplementary Information. Source data for the figures are provided with this paper. Raw data and analyzed datasets for all chips, cells, mice, and human samples generated for this study are too large to be publicly shared, yet they are available from the corresponding author on request. Responses can be expected within four weeks. Source data are provided with this paper.

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

## Acknowledgements

We would like to acknowledge the cleanroom in St. Michael's Hospital and the CRAFT cleanroom at the University of Toronto. for providing facilities, materials, and consumables for microfabrication. The reported study is supported by the Nanomedicines Innovation Network (NMIN) of Canada (Project Number: 2019-T3-02 to S.O.K.).

## Author contributions

K.C., B.T.V.D., and S.O.K. conceived the concepts and designed the experiments. M.L. and J.D. revised the experiment design. K.C., B.T.V.D, S.U.A, P.D., Z.W, C.F., J.X., Y.Z., H.W., and X.L. performed the experiments. K.C. and B.T.V.D. analyzed the experimental data. K.C. and H.Z. did the numerical simulation. Y.M. gave critical comments and revised the device design. K.C. and B.T.V.D. drafted the manuscript. All authors discussed the results and contributed to the preparation and editing of the manuscript.

## Competing interests

The authors declare no competing interests.
