## [Peer Review File · Nature Communications]

Reviewers' Comments:

Reviewer #1:

Remarks to the Author:

The manuscript by Kelley and colleagues describes a platform that is claimed to enable phenotypic sorting and exoPD-L1 profiling from blood plasma. It also shows that the exosomes with higher expression of PD-L1 have stronger immune suppression capacity on the activation of CD8+ T cells. I think this platform is interesting in light of current efforts to predict the outcome of immune therapy. However, the following issues need to be addressed to sustain their conclusions:

1. In Fig 6i, the T cells cytotoxicity assay, please indicate clearly what kind of antigen-specific T cells were used in the experiments.
2. In some experiments, the authors used the plasma spiked with cancer cell derived EVs. A number of studies showed that EV in plasma is low compared to other biological fluids, including milk and seminal plasma. How much plasma is needed in this platform?
3. In Fig 3K, the authors compared the level of PD-L1 in EVs with different levels of PD-L1 expression. Did the author load the same amount of proteins or same number of particles? It seems that L and M have similar levels of PD-L1. What is the separation resolution of this method between EV with low PD-L1 expression and EV with medium PD-L1 expression?
4. CD9 should also be included in Figure 4.
5. How efficient can the PD-L1 bead conjugates label PD-L1 on EV? If the labelling efficiency is low, the EV with low expression of PD-L1 may be inappropriately regarded as no expression of PD-L1.
6. Although the PD-L1 antibody used in this study looks specific, isotype control antibody is needed to confirm its specificity.
7. The level of PD-L1 on subtypes of EV is different as the authors show in this study. In Fig 6, the amount of same number of EV needs to be used instead of same amount of proteins.
8. The term "exosome" is used for the small extracellular vesicles (EVs) throughout the text. Not all the EVs in the 100,000 x g pellets are exosomes, which are generated from the endosomal system. There is no such a method that can completely separate exosomes from other types of small EVs because of their overlapped physical properties, such as size and density.
9. Supplementary Fig 4, the EV from H1975 seems to have larger size compared with those of PC9 and H460. It is necessary to show the particle distribution of these three cell lines. If the EVs have larger size, there should be more antigens on a single EV. Thus, the larger sized EV can be labelled much better than the EV with smaller size. Or the authors can quantify the EV size by analyzing the particle in TEM data.
10. Experiments on at least some clinical samples are needed to strengthen their claim on the potential clinical applications.
11. The TEM images with the scale bar at 100nm/50nm, and as mentioned in the Materials section, the nanoparticles used for the exosome binding is 10nm, however, the scale for the vesicles (EVs mostly less than 150nm) and nanoparticles seems not match with the scale bar in Fig. 2d, Fig. 3g and supplementary Fig. 4.
12. The TEM images on the binding of nanoparticles to EVs shows a lot of background nanoparticles around the vesicles (Supplementary Fig 4). The non-binding nanoparticles suggest that the washing is incomplete and may raise the level of non-specificity.
13. According to the guideline for the EVs studies, the WB figures should show at least one negative control and 2 positive control makers, Fig. 3 and Fig. S3 should provide more controls in the figures.

Reviewer #2:

Remarks to the Author:

This paper described a technology using magnetic nanoparticles to bind to exosome surface PDL-1, then based on the different level of PDL-1 expression to lead to different magnetic activated ranking to sort exosomes into three populations with low, medium, and high expression of PDL-1. It could be an interesting technology. However, for profiling the PDL-1 expression level intended for evaluating immunotherapy response, the quantitative measurements may make more sense, other than rough estimation as low, medium and high. Such exosome PDL-1 detection has been extensively studied from the past 10 years, to list some examples as below:

Molecular Cancer volume 21, Article number: 20 (2022)

Cells 2021, 10, 3247.
Nature volume 560, pages 382–386 (2018)

Therefore, the novelty is not very clear from this manuscript. Simply the ELISA or flow cytometry can do the same job with more quantitative measurement of PDL-1 expression level.

The ability to sort the exosome into different populations via different magnetic activated ranking could be useful. However, it is not convincing why the low, medium, and high level of PDL-1 exosomes would be different subpopulations. The PDL-1 positive and negative exosomes would be making more sense to be considered as the different subpopulations. Consequently, the single marker sorting may not be very helpful for defining different subpopulations of exosomes.

In Figure 6, the unsorted exosomes performed close effects as the high level of PDL-1 exosomes. The experiment should show the PDL-1 negative exosomes as a control, which may perform dramatically different or opposite to PDL-1 positive exosome subpopulations. In turn, it also indicates that sorting PDL-1 exosomes into low, medium, and high levels may not be meaningful, other than PDL-1 negative or positive populations.

In the conclusion, authors claimed about the application for therapeutic delivery via sorted exosomes. However, the throughput, capacity, and scaling up potential are not mentioned or discussed. So, it is hard to see such therapeutic delivery application potential.

Reviewer #3:

Remarks to the Author:

Chen and coauthors introduces a novel magneto-activated cytometry platform termed NanoEPIC to sort exosomes exoPD-L1 by their surface protein expression, aiming to predict anti-PD-1 immunotherapeutic responses through the interrogation of exoPD-L1. Overall, the manuscript is well-written and the motivation behind sorting and profiling is clearly described. Details of platform design, operation and experimental procedures are clearly presented. Many characterizations and experiments have been conducted to support the claims. In particular, the results of interaction of T cells and sorted exosome subpopulations bring new knowledge to the field. Therefore, I would recommend the publication of this manuscript after the authors consider and address the following concerns.

1. Despite supporting evidence from a series of characterization and experiments, the focus of experiments is somehow unclear and not centered. For example, the correlation between exoPD-L1 and cPD-L1 may not be something important considering that the purpose of developing NanoEPIC is to do the phenotypic profiling of exoPD-L1 and predict anti-PD-1 immunotherapeutic responses.
2. It is mentioned that "Unmatched by any existing technology to date, the NanoEPIC platform can conduct high-throughput and high-resolution sorting of individual exosomes based on surface marker expression." in the introduction, yet it is unclear how the authors defined and assessed the throughput of the system.
3. It is also stated that "unlabeled exosomes can escape to the exo-L outlet". Please clarify how the accuracy of efficiency for each outlet was determined? Could the correction value itself be "outlet" specific as well? Would it be possible to provide validation for the mathematical equation? Also, the deflection efficiency DE seems to be an overall efficiency for each outlet.
4. The accuracy evaluation of the chip needs to be elaborated. After you mentioned "further evaluate the accuracy of the device", you re-illustrated the effect of flow rate, which is demonstrated previously that the optimal flow rate is 200 $\mu\text{L}/\text{min}$ based on the simulation. However, that is not a metric for accuracy. Then, you demonstrated the high resolution of the chip by comparing the ability to detect/distinguish IFN- γ treated and untreated samples. However, that is also not a metric for accuracy.
5. Seems like the efficiency of your device is "86.6 \pm 2.9%", tested with anti-CD9, and 64.2 \pm 2.2% with wildtype exosomes. However, other methods used to isolate PD-L1 from plasma have been conducted and reported a capture efficiency of 96.5% (Pang et al. Personalized detection of circling exosomal PD-L1 based on Fe₃O₄@TiO₂ isolation and SERS immunoassay). Since you are promoting a cytometer, the accuracy and efficiency of the chip are critical.

6. The in-vivo analysis and comparison are good, yet more information should be provided to explain the opposite behavior found in the partial responder group.

RESPONSE TO REVIEWER COMMENTS

Reviewer #1 (Remarks to the Author):

The manuscript by Kelley and colleagues describes a platform that is claimed to enable phenotypic sorting and exoPD-L1 profiling from blood plasma. It also shows that the exosomes with higher expression of PD-L1 have stronger immune suppression capacity on the activation of CD8+ T cells. I think this platform is interesting in light of current efforts to predict the outcome of immune therapy. However, the following issues need to be addressed to sustain their conclusions:

1. In Fig 6i, the T cells cytotoxicity assay, please indicate clearly what kind of antigen-specific T cells were used in the experiments.

Response: The T cells used in Fig 6i are CD8⁺ T cells. We appreciate the reviewer pointing this out and added the following explanation in the main text.

"Through LDH assay, we observed a significant reduction in CD8+ T cell cytotoxicity when treated with sEVs with higher exoPD-L1 levels (Fig. 6i)." (Main text Page 13, Paragraph 1)

2. In some experiments, the authors used the plasma spiked with cancer cell derived EVs. A number of studies showed that EV in plasma is low compared to other biological fluids, including milk and seminal plasma. How much plasma is needed in this platform?

Response: We appreciate the review pointing out the need for clarification. According to literature, the concentration in EVs in plasma samples between 10^{11} - 10^{12} particles/mL (<https://doi.org/10.1002/jex2.46>; <https://doi.org/10.1038/s41598-020-59523-00>). We have also measured the amount of EVs in plasma from different patient samples. The general amount is around 1×10^{11} particles per mL. We used 50 μ L of patient plasma sample for exoPD-L1 profiling. We have added a section in the Methods detailing the processing of plasma samples from patients.

"Clinical Sample Processing. Plasma samples were obtained from de-identified cancer patients who were undergoing anti-PD-1 or anti-PD-L1 immunotherapy, procured from PrecisionMed LLC (Supplementary Table 4). All plasma samples were pre-filtered with 0.22 μ m filters. The concentration of sEV in the plasma was measured to be between 0.6- 1.74×10^{11} particles/mL. For sample processing, 50 μ L of plasma sample was first diluted 4 times with 1% filtered BSA. The diluted sample was then treated with 2.5 μ L of anti-PD-L1 conjugated MNPs and incubated at 4 °C overnight. The mixture was then processed through the NanoEPIC system at 200 μ A. sEVs collected from different outlets were analyzed with NTA as mentioned above." (Main text Page 20, Paragraph 5)

3. In Fig 3K, the authors compared the level of PD-L1 in EVs with different levels of PD-L1 expression. Did the author load the same amount of proteins or same number of particles? It seems that L and M have similar levels of PD-L1. What is the separation resolution of this method between EV with low PD-L1 expression and EV with medium PD-L1 expression?

Response: Thank you for the comment. As mentioned in the methods section, we used the same amount of protein for each sample analyzed by Western Blot. The separation of different EV subpopulation was detailed in the section "Calculation of the NanoEPIC score" of the SI. (SI Page 5-7) Theoretically, the PD-L1 expression of M is 1.67 times the expression of L and the PD-L1 expression of H is 3.32 times the expression of L. The WB shown in Fig. 3K is an example of a single qualitative measurement of the exoPD-L1 of L, M, H subpopulations. We also performed multiple quantitative measurements of L, M, H subpopulations using ELISA. We can see significant differences between L, M, H as shown Fig. 3j. Moreover, we have performed more WB of different sEV subpopulations as requested. Exo-M subpopulations shows significantly higher PD-L1 expression compared with Exo-L subpopulations. See Updated Fig. 3K.

4. CD9 should also be included in Figure 4.

Response: Thank you for the suggestion. We assume that the reviewer was looking for the sEV concentration in the plasma of mouse models given that CD9 is a typical biomarker for sEVs. We added the concentration of EVs from all the mice (Supplementary Fig. 10d). We also added this discussion in the main text.

"We then investigated if plasma levels of sEVs are correlated with outcomes. The concentration of sEVs from all the mice were measured using NTA. No significant differences were observed between complete responders, partial responders and non-responders (Supplementary Fig. 10d)." (Main text Page 9, Paragraph 1)

5. How efficient can the PD-L1 bead conjugates label PD-L1 on EV? If the labelling efficiency is low, the EV with low expression of PD-L1 may be inappropriately regarded as no expression of PD-L1.

Response: We appreciate the review pointing out the need for clarification of this important point. For PD-L1 bead conjugation, we treated EVs with excessive amount of PD-L1 MNPs to ensure maximum interaction between EVs and MNPs. In fact, we have compared the deflected exosomes and undeflected exosomes in terms of PD-L1 expression. Fig. 3j shows that the PD-L1 expression of undeflected exosomes are significantly lower than the deflected exosomes. Therefore, it would be safe to say the MNP conjugation efficiency is high enough for exoPD-L1 profiling. To make it clearer, we have added one sentence in the method section "Magnetic labeling of sEVs" to emphasize that excessive amount of functionalized MNPs were used to guarantee maximized conjugation efficiency.

"To ensure the maximum interaction between EVs and MNPs and maximized magnetic labeling efficiency, an excessive amount of functionalized MNPs were added to the sEV sample." (Main text Page 18, Paragraph 2)

6. Although the PD-L1 antibody used in this study looks specific, isotype control antibody is needed to confirm its specificity.

Response: Thank you for the suggestion. We added an isotype control to Fig. 1h and Fig. 1i.

7. The level of PD-L1 on subtypes of EV is different as the authors show in this study. In Fig 6, the amount of same number of EV needs to be used instead of same amount of proteins.

Response: Thank you for the suggestion. We used protein weight as it reflects the total sEV number. To address the reviewer's concern, we made a calibration curve to confirm the linear relation between sEV number and protein amount (Supplementary Fig. 6c).

8. The term "exosome" is used for the small extracellular vesicles (EVs) throughout the text. Not all the EVs in the 100.000 x g pellets are exosomes, which are generated from the endosomal system. There is no such a method that can completely separate exosomes from other types of small EVs because of their overlapped physical properties, such as size and density.

Response: Thank you for the suggestion. We changed the term "exosome" to "sEV".

9. Supplementary Fig 4, the EV from H1975 seems to have larger size compared with those of PC9 and H460. It is necessary to show the particle distribution of these three cell lines. If the EVs have larger size, there should be more antigens on a single EV. Thus, the larger sized EV can be labelled much better than the EV with smaller size. Or the authors can quantify the EV size by analyzing the particle in TEM data.

Response: We appreciate the careful comparison of the TEM images. To address this comment, we added NTA analysis of EVs of H1975, PC9 and H460 cells. As given in Supplementary Fig. 4c, we didn't observe significant differences in size distribution between different EVs. Furthermore, we added more TEM images of EVs from H1975, PC9 and H460. These images show that the number of MNPs bonded to an EV is not necessarily related to the size of the EV (Supplementary Fig. 4b).

10. Experiments on at least some clinical samples are needed to strengthen their claim on the potential clinical applications.

Response: We appreciate this feedback and accordingly processed additional samples from patients undergoing anti-PD-1/anti-PD-L1 immunotherapy and healthy volunteers. We have added an additional figure (Figure 7 and supplementary 17) to address this along with a discussion in the main text. (Main text Page 13, Paragraph 2)

11. The TEM images with the scale bar at 100nm/50nm, and as mentioned in the Materials section, the nanoparticles used for the exosome binding is 10nm, however, the scale for the vesicles (EVs mostly less than 150nm) and nanoparticles seems not match with the scale bar in Fig. 2d, Fig. 3g and supplementary Fig. 4.

Response: We appreciate the reviewer highlighting this point. For nanoparticles, the core magnetic component made of iron-oxide has a diameter of 10 nm. As described by the manufacturer, the core magnetic component is coated with an amphiphilic polymer layer conjugated to protein G, which made the hydrodynamic size of the nanoparticles about 8-10 nm larger than the magnetic core. For the size of EVs, we have added NTA analysis of EVs from H1975, PC9 and H460 (Supplementary Fig. 4c). Also, we added new TEM images as

mentioned above.

12. The TEM images on the binding of nanoparticles to EVs shows a lot of background nanoparticles around the vesicles (Supplementary Fig 4). The non-binding nanoparticles suggest that the washing is incomplete and may raise the level of non-specificity.

Response: Thank you for the comment. TEM images acquired for Supplementary Fig. 4 were used for observation of MNP binding, and they have not been processed through the NanoEPIC. The excess numbers of MNPs were used to ensure optimal binding efficiency. The effect of excessive MNPs can be eliminated through gating during NTA analysis. Furthermore, the sorted EVs can be further purified through precipitation if necessary. As for the non-specific binding of MNPs on the EV, we have shown isotype control and negative control in Fig. 1h and 1i. Therefore, it would be safe to say that the background MNPs will not significantly affect the performance of the NanoEPIC system in terms of specificity.

13. According to the guideline for the EVs studies, the WB figures should show at least one negative control and 2 positive control makers, Fig. 3 and Fig. S3 should provide more controls in the figures.

Response: We performed WB again with 2 positive control markers and one negative control marker as suggested. See updated Fig. 3k and Fig. S3c.

Reviewer #2 (Remarks to the Author):

This paper described a technology using magnetic nanoparticles to bind to exosome surface PDL-1, then based on the different level of PDL-1 expression to lead to different magnetic activated ranking to sort exosomes into three populations with low, medium, and high expression of PDL-1. It could be an interesting technology. However, for profiling the PDL-1 expression level intended for evaluating immunotherapy response, the quantitative measurements may make more sense, other than rough estimation as low, medium and high. Such exosome PDL-1 detection has been extensively studied from the past 10 years, to list some examples as below: Molecular Cancer volume 21, Article number: 20 (2022); Cells 2021, 10, 3247 (Please add these two papers as references in the introduction); Nature volume 560, pages 382-386 (2018)

Therefore, the novelty is not very clear from this manuscript. Simply the ELISA or flow cytometry can do the same job with more quantitative measurement of PDL-1 expression level.

Response: We appreciate these constructive comments. We agree that ELISA or flow cytometry can perform quantitative measurement of exoPD-L1. As mentioned in the abstract, most of these measurements are performed on a bulk EV sample where the heterogeneity of individual EVs are largely ignored. Some high ended flow cytometry was developed or optimized for EV profiling (Ref. 18) but it still lack the throughput for up-scaled sorting of EVs based on a surface marker. The main focus of the manuscript is not detecting exoPD-L1. The development of the NanoEPIC system is to profile exoPD-L1 in individual exosomes and sort exosomes into different subpopulations based on their PD-L1 expression with a high

throughput. To address this comment, we add more content in the introduction (highlighted in yellow) to emphasize the advantages of the NanoEPIC system as follows:

"In fact, with optimized high-end cytometers, PD-L1 positive EVs have been detected from clinical samples and shown relevance to immunotherapy outcome^{18,19}. However, the detection of EVs with a size smaller than 100 nm is still a challenge and the throughput needs to be improved." (Main text Page 2, Paragraph 3)

"Surface marker profiling (e.g. EpCAM) of circulating tumor cells (CTCs) have shown clinical relevance with cancer progression"⁵. We also found that cell subpopulations with different surface markers showed distinctive behaviors. For example, tumour-infiltrating lymphocytes (TILs) with medium CD39 expression are more potent in killing cancer cells compared with TILs with high or low CD39 expression⁵¹. We then ask if the biomarker profiling of sEVs can bring a hint to therapeutic outcomes." (Main text Page 3, Paragraph 1)

The ability to sort the exosome into different populations via different magnetic activated ranking could be useful. However, it is not convincing why the low, medium, and high level of PDL-1 exosomes would be different subpopulations. The PDL-1 positive and negative exosomes would be making more sense to be considered as the different subpopulations. Consequently, the single marker sorting may not be very helpful for defining different subpopulations of exosomes.

Response: We agree with the review that PD-L1 positive and negative can be clearly considered as different subpopulations. However, simply separating "positive" and "negative" might not decipher the whole exoPD-L1 expression spectrum within the exosomes. At the cellular level, studies have shown that cell subpopulations with distinct mark expressions can show significantly different behaviors (Main text Page 3, Paragraph 1). It is reasonable to assume that such phenomenon can exist in sEVs too given that EVs with different PD-L1 expressions inhibit CD8 T cells differently (Ref. 23). One main highlight of this work is to profile exoPD-L1 at a higher resolution. As shown in Fig 3, we showed that different exosome origins displayed different profiling patterns. In Fig 6, we showed that these different subpopulations caused different immune inhibition behaviors. And the trends showed that elevated PD-L1 expression in single exosome level can increase its chance of binding to a T cell and cause immune suppression (Fig. 6a and b). In this regard, we believe that exosome subpopulations with different PD-L1 expressions can actually behave differently. Furthermore, we added a paragraph in the conclusion to discuss the potential of NanoEPIC for multiplex mark analysis and sorting (Main text Page 16, Paragraph 2).

In Figure 6, the unsorted exosomes performed close effects as the high level of PDL-1 exosomes. The experiment should show the PDL-1 negative exosomes as a control, which may perform dramatically different or opposite to PDL-1 positive exosome subpopulations. In turn, it also indicates that sorting PDL-1 exosomes into low, medium, and high levels may not be meaningful, other than PDL-1 negative or positive populations.

Response: Thank you for the constructive comment. The reviewer is correct that our downstream analysis showed that unsorted exosome showed similar effects as the exo-H exosomes for some markers (e.g. GzB). However, we did see differences between exo-H and exo-L, or exo-M and exo-L. That means the separation of distinctive sEV subpopulations with different exoPD-L1 expressions is possible. We have also performed the experiment on PD-L1

negative exosomes and have updated Figure 6 to address this. From our observations, PD-L1 negative exosomes displayed no statistical difference with no exosomal treatment (Figure 6c-e). Similarly, PD-L1 negative exosomes displayed a statistical difference between exo-L, exo-M, and exo-H populations for GzB, ki67, and CD69. As we are able to obtain more phenotypic expressions with sorted exosomes compared to just PD-L1 positive and negative exosomes (i.e. in Fig 6c-e), these findings further supports our claim that our platform has the resolution to sort and profile exosomes with biologically distinct activity towards CD8+ T cells.

In the conclusion, authors claimed about the application for therapeutic delivery via sorted exosomes. However, the throughput, capacity, and scaling up potential are not mentioned or discussed. So, it is hard to see such therapeutic delivery application potential.

Response: We appreciate the feedback. The throughput and capacity of the NanoEPIC was detailed in "**Development of the NanoEPIC platform**" section (Main text Page 6, Paragraph 2; Fig. 3 a and b). We also added several sentences in the conclusion to discuss the potential of the NanoEPIC system for therapeutic delivery.

"Given that drug loading to sEVs can also be heterogeneous⁵⁸, the NanoEPIC system can be used to sort out sEV subpopulations with high drug loading, which can potentially reduce the dose of sEVs required for effective treatment. Another application of NanoEPIC can be sorting of cargo-loaded sEVs based on their specificity to recipient cells. sEVs engineered with antibodies specific to recipient cells have been used for cargo delivery⁵⁹. Sorting of high marker expressed sEVs can increase the uptake of cargo-loaded sEVs by the recipient cells. Also, we have shown that the NanoEPIC system was able to sort magnetically labeled concentrated sEVs (up to 10^{12} particles/mL) with high deflection efficiency (more than 80%) at a flow rate of 200 pL/h. The NanoEPIC was amenable for simultaneous processing of up to 6 microfluidic chips, which further increases its total throughput." (Main text Page 16, Paragraph 1)

Reviewer #3 (Remarks to the Author):

Chen and coauthors introduces a novel magneto-activated cytometry platform termed NanoEPIC to sort exosomes exoPD-L1 by their surface protein expression, aiming to predict anti-PD-1 immunotherapeutic responses through the interrogation of exoPD-L1. Overall, the manuscript is well-written and the motivation behind sorting and profiling is clearly described. Details of platform design, operation and experimental procedures are clearly presented. Many characterizations and experiments have been conducted to support the claims. In particular, the results of interaction of T cells and sorted exosome subpopulations bring new knowledge to the field. Therefore, I would recommend the publication of this manuscript after the authors consider and address the following concerns.

1. Despite supporting evidence from a series of characterization and experiments, the focus of experiments is somehow unclear and not centered. For example, the correlation between exoPD-L1 and cPD-L1 may not be something important considering that the purpose of developing NanoEPIC is to do the phenotypic profiling of exoPD-L1 and predict anti-PD-1 immunotherapeutic responses.

Response: Thank you very much for the constructive comment. We agreed with the reviewer that the focus of this work was to describe the development of NanoEPIC for exoPD-L1 profiling which can be usable for the prediction of immunotherapy responses. Since cPD-L1 is the golden standard used to determine PD-L1 levels in a patient, the correlation between exoPD-L1 and cPD-L1 provides a reliable comparison for clinicians and further supports the predictability of exoPD-L1 profiling for immunotherapeutic monitoring. The main advantage of this result is that we can obtain meaningful PD-L1 expression data using simple and minimally invasiveness protocols, which is often a great challenge in traditional methods for PD-L1 cellular evaluations such as IHC.

2. It is mentioned that "Unmatched by any existing technology to date, the NanoEPIC platform can conduct high-throughput and high-resolution sorting of individual exosomes based on surface marker expression." in the introduction, yet it is unclear how the authors defined and assessed the throughput of the system.

Response: The throughput here includes two aspects: 1) a fast sample process flow rate ; 2) the ability to process highly concentrated EV samples (e.g. 10^9 to 10^{12} EVs/mL). We have adjusted the wording to improve the clarity. (Main text Main text Page 16, Paragraph 1)

3. It is also stated that "unlabeled exosomes can escape to the exo-L outlet". Please clarify how the accuracy of efficiency for each outlet was determined? Could the correction value itself be "outlet" specific as well? Would it be possible to provide validation for the mathematical equation? Also, the deflection efficiency DE seems to be an overall efficiency for each outlet.

Response: We appreciate the request for clarification. The deflection efficiency (DE) is defined as the number of total deflected exosomes (including the exosomes collected from L, M, H outlets) over the number of total exosomes introduced to the NanoEPIC system. The profile distribution is defined as the ratios of exosomes collected from a certain outlet (L, M, or H) over the total deflected exosomes. To address this comment, we have added more description in the section of NanoEPIC score calculation in the supplemental materials.

"The DE is defined as the number of total deflected sEVs (including the sEVs collected from L, M, H outlets) over the number of total sEVs introduced to the NanoEPIC system. The profile distribution is defined as the ratios of sEVs collected from a certain outlet (L, M, or H) over the total deflected sEVs." (Supplementary Information Page 5)

4. The accuracy evaluation of the chip needs to be elaborated. After you mentioned "further evaluate the accuracy of the device", you re-illustrated the effect of flow rate, which is demonstrated previously that the optimal flow rate is 200 μ L/min based on the simulation. However, that is not a metric for accuracy. Then, you demonstrated the high resolution of the chip by comparing the ability to detect distinguish IFN- γ treated and untreated samples. However, that is also not a metric for accuracy.

Response: Thank you for the constructive feedback comment. The main purpose of these tests was to demonstrate the performance of NanoEPIC in exoPD-L1 profiling. We agree with the reviewer that using "accuracy" is not an accurate description of the tests. Therefore, we changed it to "performance". (Main text Page 8, Paragraph 1)

5. Seems like the efficiency of your device is "86.6 \pm 2.9%", tested with anti-CD9, and 64.2 \pm 2.2%

with wildtype exosomes. However, other methods used to isolate PD-L1 from plasma have been conducted and reported a capture efficiency of 96.5% (Pang et al. Personalized detection of circling exosomal PD-L1 based on Fe₃O₄@TiO₂ isolation and SERS immunoassay). Since you are promoting a cytometer, the accuracy and efficiency of the chip are critical.

Response: Thank you for this constructive feedback. We agree with the reviewer that the accuracy and efficiency of the chip are critical. As mentioned above, the deflection efficiency in this work is defined as the ratio of deflected exosomes (PD-L1 positive exosomes) and the total number of exosomes. In other words, deflection efficiency here represents the proportion of exosomes that are PD-L1+. While 64.2±2.2% with wildtype exosomes are deflected, exoPD-L1 of undeflected exosomes is extremely low as shown in Fig. 3j, showing a high sensitivity of the NanoEPIC system. Also, the deflection efficiency of PD-L1 KO exosomes and isotype control was only 5.4% and 8.6% respectively (Fig. 3j), implying a low nonspecific deflection of exosomes.

Combining the results of adequate deflection of PD-L1+ exosomes and low deflection of PD-L1- exosomes implies that our technology is quite accurate, but since we are unable to obtain a true positive / negative value (i.e. known total amount of PD-L1+ and PD-L1- exosomes) due to the limitation of current technology, we are unable to define accuracy further and can only support our claims as we have. Similarly, the ability to sort a high number of exosomes with high throughput and low non-specific sorting implies high sorting efficiency.

We also appreciate the reference to the work by Pang et al. showing superb capture efficiency of exosomes. However, the efficiency here is defined differently. Since the capture of exosomes is based on the hydrophilic phosphate head, not a specific surface marker, the capture efficiency in this work is the ratio of pooled down exosomes to the total exosomes. The captured exosomes are not necessarily PD-L1+ exosomes.

6. The in-vivo analysis and comparison are good, yet more information should be provided to explain the opposite behavior found in the partial responder group.

Response: While we appreciate the feedback, we observe that partial responders do not necessarily have the opposite response compared to non-responders (although it may look like it in Fig 4b heatmap). More exactly, the partial responder group contains a wide diversity of PD-L1 expressions while the non-responders are essentially saturated in exo-PD-L1 expressions. We further elaborate this spectrum of PD-L1 expressions in figure 4c, where tumor volume increases more quickly at lower NanoEPIC scores and saturates at high scores.

Reviewers' Comments:

Reviewer #1:

Remarks to the Author:

The authors addressed most of my questions. I support publication.

Reviewer #2:

None

Reviewer #3:

Remarks to the Author:

The authors have addressed all my previous concerns. I recommend the publication.